# Orai inhibition modulates pulmonary ILC2 metabolism and alleviates airway hyperreactivity in murine and humanized models

Emily Howard[1], Benjamin P. Hurrell[1], Doumet Georges Helou [1],
Pedram Shafiei-Jahani[1], Spyridon Hasiakos[2], Jacob Painter[1], Sonal Srikanth [2],
Yousang Gwack [2] & Omid Akbari [1] ✉

$Ca^{2+}$ entry via $Ca^{2+}$ release-activated $Ca^{2+}$ (CRAC) channels is a predominant mechanism of intracellular $Ca^{2+}$ elevation in immune cells. Here we show the immunoregulatory role of CRAC channel components Orai1 and Orai2 in Group 2 innate lymphoid cells (ILC2s), that play crucial roles in the induction of type 2 inflammation. We find that blocking or genetic ablation of Orai1 and Orai2 downregulates ILC2 effector function and cytokine production, consequently ameliorating the development of ILC2-mediated airway inflammation in multiple murine models. Mechanistically, ILC2 metabolic and mitochondrial homeostasis are inhibited and lead to the upregulation of reactive oxygen species production. We confirm our findings in human ILC2s, as blocking Orai1 and Orai2 prevents the development of airway hyperreactivity in humanized mice. Our findings have a broad impact on the basic understanding of $Ca^{2+}$ signaling in ILC2 biology, providing potential insights into the development of therapies for the treatment of allergic and atopic inflammatory diseases.

Group 2 innate lymphoid cells (ILC2s) are innate immune cells that are rapidly activated by epithelium alarmins, namely thymic stromal lymphopoietin (TSLP), interleukin (IL)-25 and IL-33[1]. ILC2s are the dominant innate lymphoid cell population in the lungs at steady state and upon activation their release of type-2 cytokines, predominately IL-5 and IL-13, is a central driver in responding eosinophil infiltration and the development of airway hyperreactivity independent of adaptive immunity[2]. Their non-redundant role in the lung is uniquely pro-inflammatory due to their positional niche after allergen challenge. At steady state, ILC2s occupy the peripheral and central sites of the lung, but cluster at the airway epithelium and alveolar space upon airway inflammation[3]. This allows the cells to be the first responders to allergens that enter the system, driving eosinophilia, increased goblet cell activation and mucus production[4].

Due to their crucial role in the development of inflammation associated with allergic asthma, delineating the intracellular signaling and cellular mechanisms that drive ILC2 effector functions is critical for potential modulation and immunoregulatory control[5]. We and others have shown that CRAC ($Ca^{2+}$ release-activated $Ca^{2+}$) channels represent a primary mode of extracellular $Ca^{2+}$ entry in a variety of immune cells including T cells and mast cells that play an essential role in immune cell proliferation, activation, and basic effector function[6–10]. Three Orai proteins have been previously described as major subunits forming the CRAC channels[11]. Recently, it has been shown that Orai1 and Orai2 can form heterotrimeric CRAC channels, and that inhibition of both subunits will severely affect neutrophil and mast cell function[11,12]. During an immune response, depletion of the endoplasmic reticulum (ER) $Ca^{2+}$ stores activates ER-resident protein

[1]Department of Molecular Microbiology and Immunology, Keck School of Medicine, University of Southern California, Los Angeles, CA, USA. [2]Department of Physiology, David Geffen School of Medicine, University of California, Los Angeles, CA, USA. ✉e-mail: akbari@usc.edu

Stromal Interaction Molecule 1 (STIM1) to gate Orai channels on the plasma membrane, allowing for the initiation of the store-operated Ca²⁺ entry (SOCE), a major mechanism utilized by non-excitable cells to raise intracellular Ca²⁺ concentration[13]. Despite the well documented importance of CRAC channels in immune effector function, the mechanisms by which they are regulated and their role in pathogenic ILC2s, specifically in the context of the development of allergic airway inflammation remains unresolved.

Recently, our group and others have detailed the importance of metabolism and mitochondrial health in ILC2 effector function[14–16]. It has been previously demonstrated that cytokine IL-5 and IL-13 secretion by pathogenic ILC2s is dictated by the utilization of fatty acid oxidation (FAO) to generate ATP[14,15]. In the absence of FA or if the pathway is defective in a way that limits ATP generation, ILC2s will metabolically adapt to utilize glycolysis[15]. Emerging evidence demonstrates that Ca²⁺ entry via the CRAC channels may play a vital role in metabolism. Impairments in Ca²⁺ exchange have been shown to lead to high alterations in the metabolism of neurons in models of Alzheimer's Disease[17]. Previously it has been shown in CD4 + Th17 cells that deletion of STIM1 and STIM2 severely affected mitochondrial and metabolic health, resulting in altered inner mitochondrial membrane architecture and reduced immune cell effector function[18]. Vaeth et al. recently documented the necessity of STIM1 for functional oxidative phosphorylation (OXPHOS) and glycolysis in non-pathogenic Th17 cells, while pathogenic Th17 cells operate glycolysis independent of Orai or CRAC channels[6]. While it is clear that metabolism has a critical role in ILC2 activation, currently the relationship between Ca²⁺ entry in ILC2s and metabolism, if any, has yet to be explored.

In this study, we show that Orai1 and Orai2 are expressed on ILC2s, and that inhibition of the channels result in an amelioration of the development of humanized ILC2-dependent airway hyperreactivity (AHR). We demonstrate that blocking the Orai channels downregulate both the FAO/OXPHOS and glycolysis in murine proinflammatory ILC2s. Further the mitochondrion is significantly affected, resulting in a downregulation of the mitochondrial electron transport chain and an upregulation in reactive oxygen species (ROS) through an uncoupling of the mitochondrial electron transport chain. We confirmed our findings in human ILC2s, as blocking Orai1 and Orai2 prevented the development of airway hyperreactivity in humanized mice. Together, our findings give further insight into the fundamental biology of ILC2s, opening avenues into potential downstream targets that may be pharmacologically manipulated to specifically modulate human ILC2s for the treatment of a broad array of inflammatory diseases. Specifically, our study can have a direct translational impact on development of improved therapies for allergic asthma.

## Results

### Calcium channels Orai1 and Orai2 are expressed on murine pulmonary ILC2s and are upregulated by IL-33 activation

The study of calcium channels in autoimmune and allergic diseases has recently gained therapeutic interest. Here we have studied the pro-inflammatory role of calcium channels Orai1 and Orai2 in eosinophilic asthma mouse models. The role of the Ca²⁺-calcineurin-NFAT pathway is well defined in immune cells, but the mechanism on how Ca²⁺ ions enter cells and its downstream effect on the cell health and survival has only recently begun to be understood. It has been previously demonstrated that immune cells largely express Orai1, and that mucosal tissues including the lung tend to express high levels of Orai2[12]. To investigate whether pulmonary ILC2s express the Orai channel isoforms, we first performed RNA-sequencing on FACs-sorted ILC2s activated with recombinant mouse (rm)IL-33 (Fig. 1A). Orai1 and Orai2 are highly expressed in activated murine ILC2s, while Orai3 demonstrated basal expression comparatively (Fig. 1A). We then took advantage of publicly available single-cell RNA sequencing (scRNA-

seq) for naive vs IL-33 activated pulmonary ILC2s to further investigate the individual cell expression of Orai1 and Orai2 (Fig. 1B). Gene counts for Orai1 and Orai2 isoforms were elevated in both naive and IL-33 induced ILC2s, suggesting these channels may play an important role in Ca²⁺ entry in pulmonary ILC2s. To explore the expression of the channel in vivo, we challenged mice for three consecutive days with recombinant mouse (rm)IL-33 and assessed expression level of the proteins in pulmonary ILC2s by flow cytometry on day four. Our data demonstrate that activated pulmonary ILC2s express significant levels of Orai1 and Orai2 (Fig. 1C, D). Interestingly, naive ILC2s expressed basal levels of Orai1 and Orai2, though IL-33 activation further enhanced the expression, especially of Orai1. The results indicated that Orai1 and Orai2 are expressed on activated pulmonary ILC2s and may play a role in the development of airway hyperreactivity.

### Blocking Orai channels on pulmonary ILC2s downregulates pro-inflammatory effector function

In order to determine the role of the expressed Orai channels in ILC2 effector function, we next aimed at determining the most efficient and specific pan-Orai inhibitor on ILC2s. Previously, using a chemical library screen, we identified compound 5D, N-[2,2,2-trichloro-1-(2-naphthylamino)ethyl]−2-furamide, that effectively blocks all Orai channels indiscriminately with an IC50 value of 195 nM in blocking SOCE in primary T cells[19]. Biologically, we have previously shown that 5D efficiently suppressed effector T-cell function and IL-2 production, suggesting that it is a potent tool to study the function of Orai channels in the context of functional immune cells[19]. Given the essential role of ILC2s in the pathology of allergic asthma and their dynamic expression of Orai channels, we examined the effect of 5D on ILC2-mediated airway inflammation. To ensure 5D does indeed block Ca²⁺ entry in pulmonary ILC2s, activated ILC2s were FACS-sorted and cultured with survival cytokines recombinant mouse (rm)IL-2 and IL-7 overnight. After 18 h, cells were loaded with Fluo-8 AM and pretreated with thapsigargin to deplete the intracellular Ca²⁺ stores. Fluorescence of the Ca²⁺ entry detected by the stain was then measured by flow cytometry before and after calcium and the inhibitor 5D addition. SOCE measurement with and without addition of compound 5D showed a strong reduction in the presence of compound 5D (Fig. 2A). To further investigate the effect of blocking Ca²⁺ in ILC2s, we examined the function of ILC2s after treatment with compound 5D in vitro (Fig. 2B). Briefly, pulmonary naive ILC2s were FACS-sorted and cultured in cRPMI for 24 h with 5D or the vehicle. After 24 h, IL-33 or saline were added to the culture for an additional 48 h. Titration of compound 5D determined that 5μM was the most effective dose with the lowest effect on viability (Figure S2A). ILC2s cultured with 5D demonstrated lower levels of proliferation marker Ki67 (Fig. 2C), as well as reduction in secreted proinflammatory cytokines, including IL-5 and IL-13 (Figs. 2D, S2B) compared to the vehicle control. Intracellular IL-5 and IL-13 levels were also lower in cells cultured with 5D, further suggesting an important role of SOCE in ILC2 pro-inflammatory effector function (Fig. 2E, Figure S2C). To confirm our findings, we generated Orai1ᶠˡ/ᶠˡ; UBC-Cre/ERT2 and Orai1ᶠˡ/ᶠˡ; Orai2⁻/⁻; UBC-Cre/ERT2 mice as described in the Methods and validation was performed to confirm the Orai2 deletion (Figure S1). We first sorted naive pulmonary ILC2s from control, Orai1ᶠˡ/ᶠˡ; UBC-Cre/ERT2 and Orai1ᶠˡ/ᶠˡ; Orai2⁻/⁻; UBC-Cre/ERT2 mice. Animals of all genotypes had similar number of ILC2s in the lungs at steady state (Figure S2D). Cells were then cultured with IL-33 to induce expansion. Further, cells were treated with 4-hydroxy tamoxifen (4-OHT) in vitro, to induce Orai1 deletion. After expansion, supernatant and intracellular cytokines were measured by ELISA and flow cytometry respectively (Fig. 2F, G). And indeed, major effector function cytokines secreted from ILC2s, including IL-5, IL-13, as well as IL-9, IL-6 are downregulated by the deletion of Orai channels (Figs. 2F, S2E). Orai1 deletion further downregulates cytokine production, though deletion of both Orai

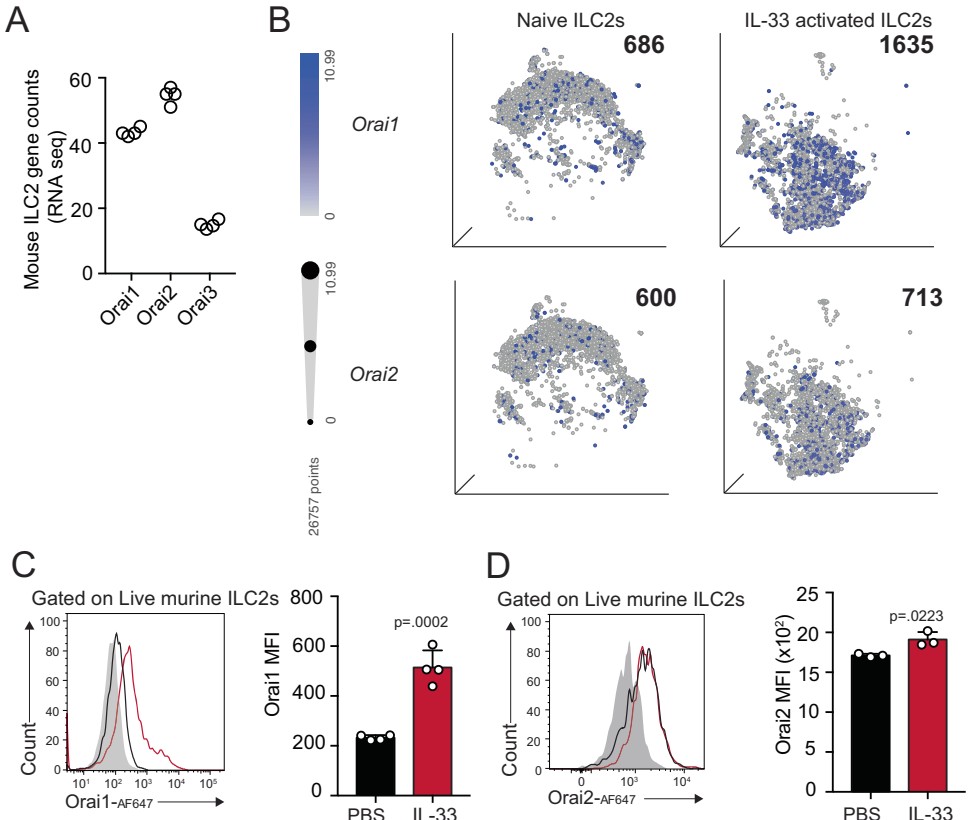

**Fig. 1 | Murine pulmonary ILC2s express calcium release activated calcium channels Orai1 and Orai2. A** *Orai1, Orai2, Orai3* expression in FACs-sorted IL-33 activated pulmonary murine ILC2s by bulk RNA-sequencing. **B** *Orai1* and *Orai2* gene expression level in naive and IL-33 activated ILC2s by single-cell RNA sequencing. Bolded number represents the quantification of the number of expressing ILC2s (blue dots) in each plot. **C** Representative flow cytometry histogram of Orai1 expression on murine ILC2s with and without IL-33 stimulation in vivo and

corresponding quantification presented as mean ± SEM. $n$ = 4 biologically independent samples. **D** Representative flow cytometry histogram of Orai2 expression on murine ILC2s with and without IL-33 stimulation in vivo and corresponding quantification presented as mean ± SEM. $n$ = 3 biologically independent samples. Data are representative of two independent experiments and are presented as means ± SEM. Source data are provided as a Source data file. A two-tailed Student's $t$ test for unpaired data were applied for comparisons between two groups.

channels has a greater effect both at the supernatant level and intracellularly (Figs. 2F, G, S2F). Proliferation is also decreased after deletion, while viability is only slightly affected after tamoxifen addition (Figure S2G, H). Together, our findings provide strong evidence that blocking $Ca^{2+}$ entry specifically through Orai channels play a crucial and previously underrecognized role in the development of effector function of ILC2s.

### Inhibiting Orai channels affects the transcriptomic landscape of pro-inflammatory pulmonary ILC2s
To further explore the relationship of ILC2s and the role Orai channels play in their effector function, we performed RNA sequencing on activated ILC2s cultured with and without 5D for 24 h. Not surprisingly, a significant number of genes were differentially expressed between vehicle-treated and compound 5D-treated cells (Fig. 3A). 160 genes were significantly ($p < 0.05$) upregulated, while 199 genes were significantly ($p < 0.05$) downregulated, totaling 359 genes differentially expressed in ILC2s treated with 5D as compared to the control. We further analyzed the pathways of differentially expressed genes by Ingenuity Pathway Analysis (IPA) (Fig. 3B). In line with our previous data, the most downregulated pathways in ILC2s cultured with 5D include Th2 pathway and activation, as well as metabolic pathways. Cholesterol biosynthesis and glycolysis appear to be significantly ($p < 0.05$) downregulated in 5D-treated ILC2s, suggesting blocking Orai channels may largely affect the mitochondria and metabolic health of the ILC2s[20]. Importantly, we analyzed the RNA sequencing (RNA-seq) data to focus specifically on mRNA related to ILC2 health and effector function. We

found that *Il9, Mki67, Il5,* and *Il6* were downregulated in ILC2s treated with 5D in confirmation that cytokines are downregulated without Orai channels (Fig. 3C, D). *Il13* was also downregulated, though not statistically. Interestingly, transcription factors important in ILC2 homeostasis were largely unaffected by Orai inhibition, suggesting an alternative mechanism by which 5D regulates cytokine production (Fig. 3D). To confirm the glycolytic pathway was downregulated as identified by IPA, we investigated the individual genes involved and indeed, glycolysis is largely decreased in Orai-inhibited ILC2s (Fig. 3E). Altogether, our data demonstrate that Orai channels support and improve ILC2 pro-inflammatory cytokine production and effector function.

### Inhibiting Orai channels downregulates metabolic pathways in pulmonary ILC2s
Orai channels and calcium entry has previously been shown to play a crucial role in metabolic processes in proinflammatory Th17 cells[18]. Our group and others have previously established that ILC2s primarily rely on fatty acid oxidation for their cytokine production and effector function[14,15]. Though we saw little significant difference in our RNA-seq data suggesting a transcriptomic difference in fatty acid oxidation, the decrease in cytokine production witnessed by ILC2s cultured with 5D led us to investigate the functional use of the fatty acid oxidation pathway in Orai-inhibited cells. We first measured fatty acid uptake in ILC2s cultured with 5D or the control vehicle (Fig. 4A). Interestingly, we found that 5D-treated ILC2s showed decreased mean fluorescence intensity of BODIPY, suggesting that fatty acid uptake is inhibited by Orai channel blocking ex vivo (Fig. 4A). To confirm the differential fatty

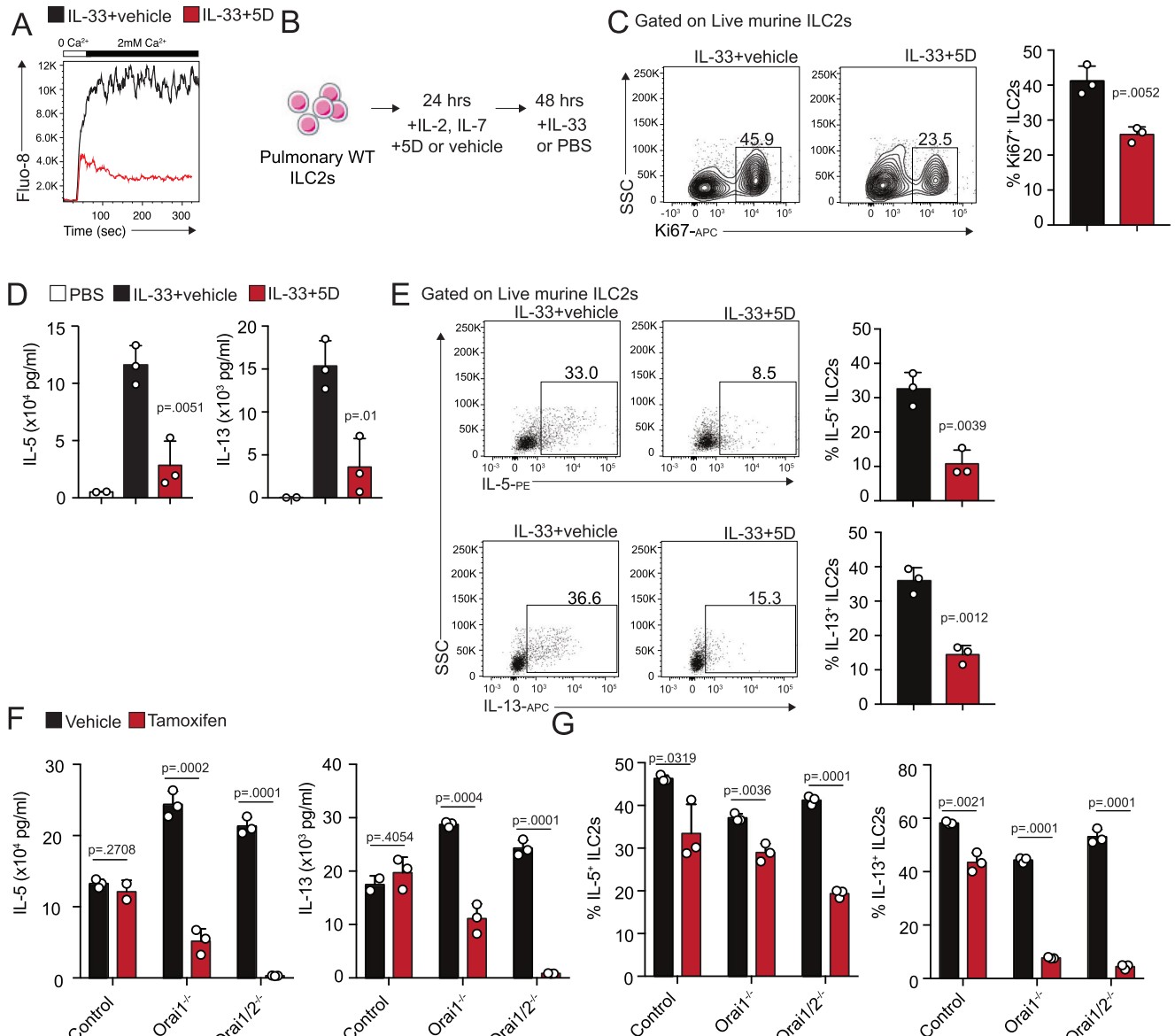

**Fig. 2 | Pulmonary ILC2 effector function is dependent on Orai1 and Orai2.**
**A** Block of endogenous SOCE in FACS-sorted pulmonary ILC2s after exposure to compound 5D measured by flow cytometry. Cells were pretreated with thapsigargin to deplete the intracellular $Ca^{2+}$ stores before addition of $Ca^{2+}$ and 5D exposure. **B** FACS-sorted pulmonary murine ILC2s were cultured for 24 h with survival cytokines rmIL-2, rmIL-7 and with rmIL-33 or PBS control for 24 h. 5D or the vehicle control were then added to the wells for an additional 48 h. **C** After 48 h, cells were collected and Ki67 as a measure of proliferation was assessed and quantified by flow cytometry presented as mean ± SEM. $n$ = 3 biologically independent samples. **D** Supernatant was collected and secreted cytokines were measured by Legendplex (S2A). Presented as mean ± SEM. $n$ = 3 biologically independent samples. **E** Representative flow cytometry plots of IL-5+ and IL-13+ ILC2s from ILC2s cultured with 5D or the vehicle and corresponding quantitation

presented as mean ± SEM. $n$ = 3 biologically independent samples. **F** Pulmonary ILC2s were sorted from wild-type (WT), $Orai1^{-/-}$ and $Orai1/2^{-/-}$ mice and cultured with and without tamoxifen for 48 h. Cells were collected, counted, and cultured for an additional 48 h. Supernatant was collected and proinflammatory cytokines were measured by Legendplex (S2B). Presented as mean ± SEM. $n$ = 3 biologically independent samples. **G** ILC2s were collected, and intracellular IL-5 and IL-13 production was measured and quantified by flow cytometry (S2C). Presented as mean ± SEM. $n$ = 3 biologically independent samples. Data are representative of 2 independent experiments and are presented as means ± SEM. Source data are provided as a Source data file. A two-tailed Student's $t$ test for unpaired data was applied for comparisons between two groups except for multi-group comparisons where Tukey's multiple comparison one-way ANOVA tests were used. ILC2 cell image designed with Servier Medical Art.

acid uptake and to understand how the metabolomics results functionally, we measured oxygen consumption rate (OCR), reflective of active oxidative phosphorylation (Fig. 4B). ILC2s treated with the Orai inhibitor demonstrated lower basal respiration (Fig. 4C), maximum respiration (Fig. 4D), and spare respiratory capacity (Fig. 4E), demonstrating that inhibition of the Orai channels has a significant detrimental effect on the mitochondrial pathways and function. Previously, our group has demonstrated that in the absence of FAO, ILC2s preferentially switch to glycolysis as an energy source[16]. However, our

previous RNA-seq data as demonstrated in Fig. 3B suggested that glycolysis is also downregulated in compound 5D-treated ILC2s. And indeed, 2-NBDG levels, as well as the L-lactate in the supernatant of cultured compound 5D-treated ILC2s versus the control were both downregulated when compared to controls (Fig. 4F, G). To confirm these results, we measured glycoPER, the glycolytic capacity of the cells utilizing Seahorse technology. As expected, compound 5D-treated ILC2s demonstrated less capacity to perform effective glycolysis than the control activated ILC2s (Fig. 4H, I). Together, these

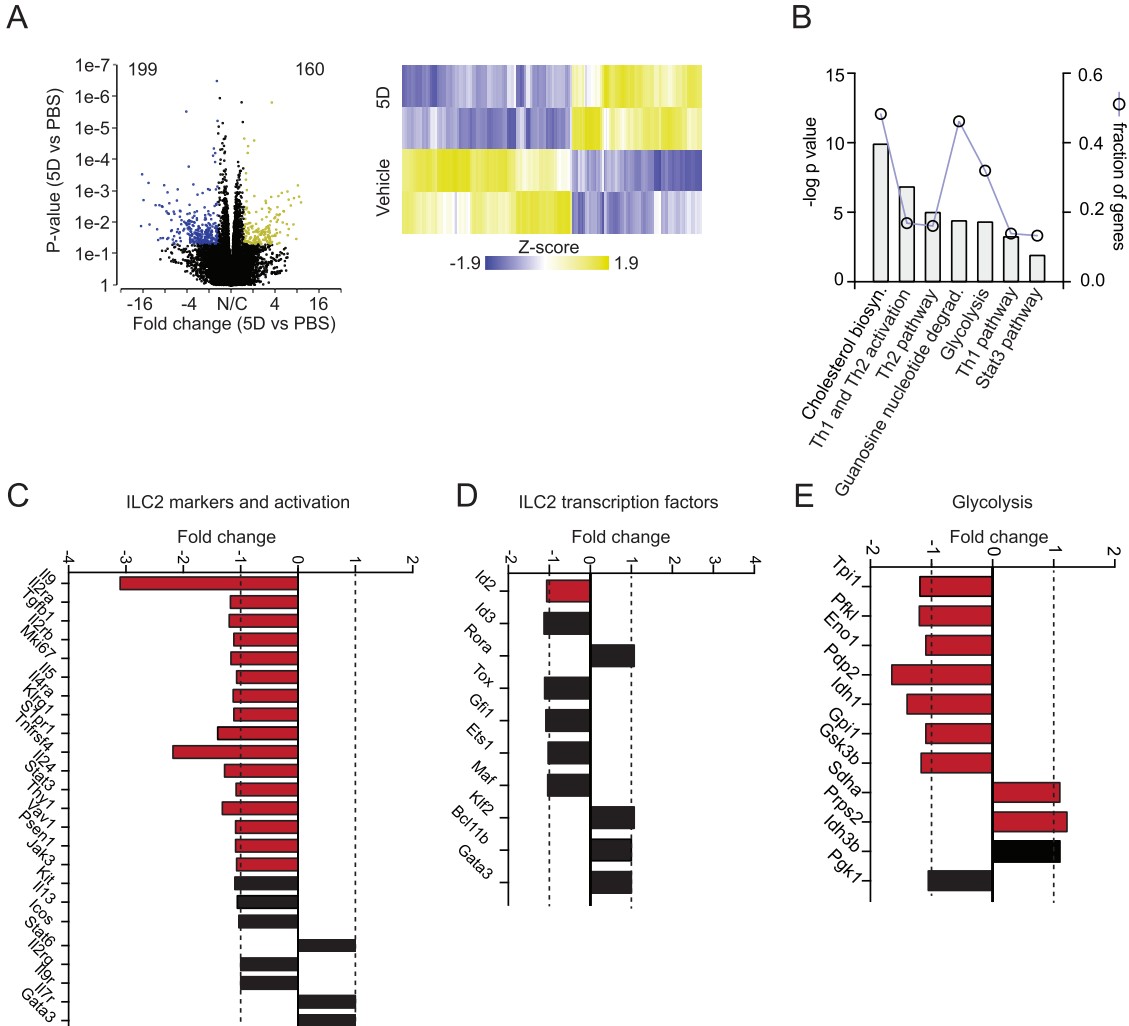

**Fig. 3 | Orai inhibition alters pro-inflammatory transcriptomic profile in pulmonary murine ILC2s. A** Volcano plot comparison of whole transcriptome gene expression of sorted ILC2s cultured with 5D or the vehicle control. Differentially expressed genes (defined as statistically significant adjusted $p$-value < 0.05) with changes of at least 2.0 fold-change (FC) are shown in yellow (upregulated genes) or blue (downregulated genes). Heat plot of selected differentially expressed genes also included. Bulk RNAseq analysis was performed by Gene Set Analysis (GSA). **B** Ingenuity Pathway Analysis (IPA) identifies pathways highly likely to be downregulated by exposure to compound 5D. The −log $p$ value is shown on the y-axis of the bar chart, and the fraction of genes identified in the pathway that are differentially expressed between our conditions, represented by the line graph right y-axis. **C** Fold change of genes involved in ILC2 markers and activation in cells exposed to 5D as compared to the control. Statistically significant differentially expressed genes are in red. **D** Fold change of genes involved in ILC2 transcription factors in cells exposed to 5D as compared to the control. Statistically significant differentially expressed genes are in red. **E** Fold change of genes involved glycolysis in ILC2s exposed to 5D as compared to the control. Statistically significant differentially expressed genes are in red. Source data are provided as a Source data file.

results demonstrate calcium influx is indispensable for metabolism, and suggests a possible defect in overall mitochondrial function after Orai inhibition.

**Orai-dependent mitochondrial function is essential for ILC2 effector function**

RNA-sequencing data, together with impaired metabolic pathways, point to a significant difference in mitochondrial function after inhibition of Orai channels. It's been demonstrated previously that functional mitochondria are critical for metabolism, both aerobic OXPHOS/FAO and anaerobic glycolysis[21,22]. Strikingly, our RNA-seq revealed a severe decrease in genes associated with various protein complexes involved in the mitochondrial electron transport chain (mtETC) when ILC2s were cultured with 5D (Fig. 5A, B). Mitochondrial mass and respiration were measured by flow cytometry as mean fluorescence of Mitotracker green and red, respectively (Fig. 5C, D), and showed a significant decrease in ILC2s cultured with 5D, further

suggesting a functional decrease in the mitochondria of the treated cells. Surprisingly, both total and mitochondrial reactive oxygen species (ROS) levels were elevated in Orai-inhibited ILC2s (Fig. 5E, F). Traditionally a lowered membrane potential coincides with lower ROS production. However it has been shown that oxidative stress can occur at very high or low levels of membrane potential and that an increase in ROS in these situations could point to mitochondrial uncoupling of the ETC[23]. To investigate whether the rise we see in ROS is the result of mitochondrial uncoupling, we measured the ratio of NADH/NAD+ in cells treated with 5D (Fig. 5G). We found treated ILC2s did have higher ratios of NADH/NAD+ while producing less ATP through oxidative phosphorylation than cells treated with the vehicle (Fig. 5H), suggesting the cells are indeed undergoing mitochondrial uncoupling. The elevated ROS resulted from this uncoupling therefore likely points to a defect in the cell's antioxidants. To investigate this, as well as whether the increased ROS could explain the loss of effector function after Orai inhibition in vitro, we utilized the addition of antioxidant

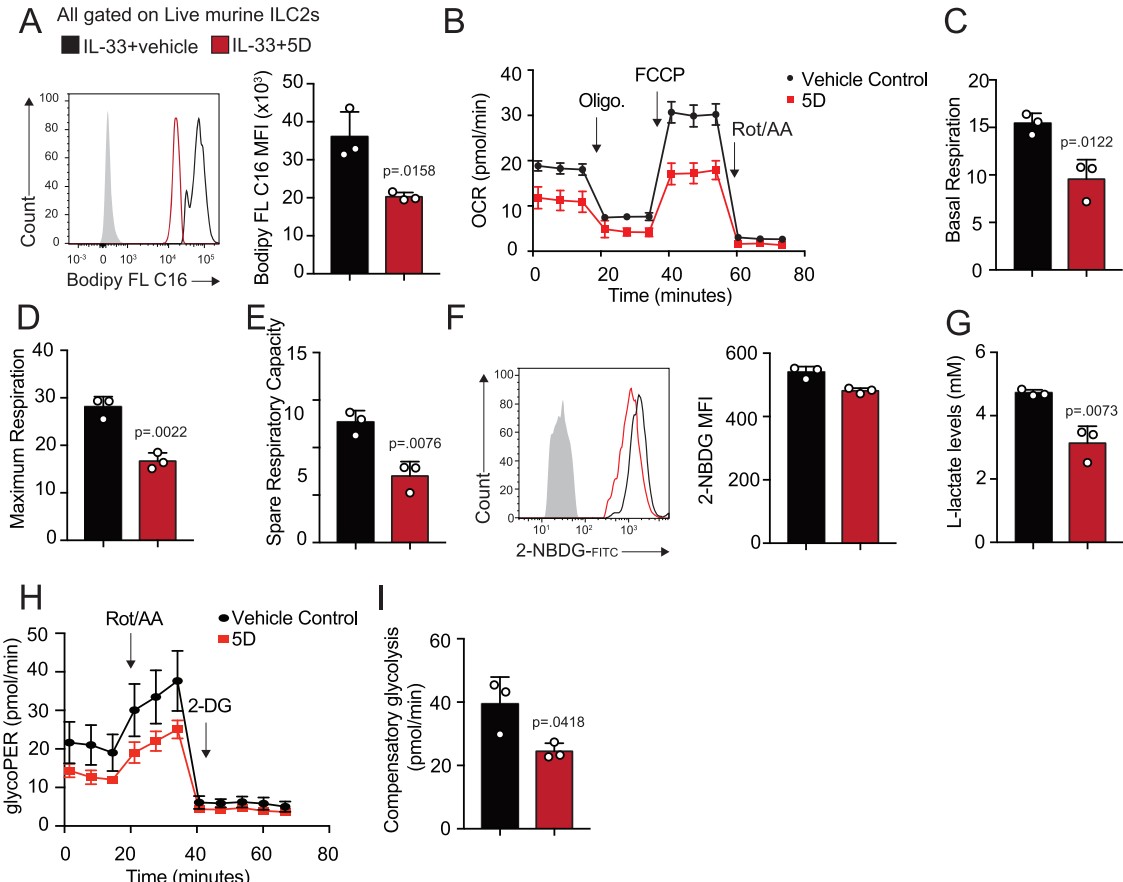

**Fig. 4 | Orai channels affect metabolic function in pulmonary murine ILC2s.**
**A** ILC2s were cultured with 5D or the vehicle. Lipid droplet uptake was measured by Bodipy FL C16 staining and quantified presented as mean ± SEM. $n$ = 3 biologically independent samples. **B** OCR was measured under basal conditions and in response to indicated drugs in 5D-treated or untreated ILC2s. $n$ = 3 biologically independent samples, presented as mean ± SEM. **C** Basal respiration as a difference in OCR presented as mean ± SEM. $n$ = 3 biologically independent samples. **D** Maximum respiration as a difference in OCR after FCCP treatment presented as mean ± SEM. $n$ = 3 biologically independent samples. **E** Spare respiratory capacity presented as the difference in OCR after FCCP treatment and basal respiration presented as mean ± SEM. $n$ = 3 biologically independent samples. **F** Treated and untreated ILC2s were generated as previously described. Glucose uptake was measured by

2-NBDG staining and quantified as 2-NBDG MFI presented as mean ± SEM. $n$ = 3 biologically independent samples. **G** Enzymatic quantification of lactate accumulation in the supernatants of cultured cells from (**F**) presented as mean ± SEM. $n$ = 3 biologically independent samples. **H** Glycolytic capacity in ILC2s cultured with 5D or the vehicle was measured in response to the indicted drugs and quantified as glycoPER. $n$ = 3 biologically independent samples, presented as mean ± SEM.
**I** Compensatory glycolysis ILC2s cultured with 5D or the vehicle was measured in response to the indicted drugs presented as mean ± SEM. $n$ = 3 biologically independent samples. Data are representative of two independent experiments and are presented as means ± SEM. Source data are provided as a Source data file. A two-tailed Student's $t$ test for unpaired data was applied for comparisons between two groups.

N-Acetyl-L-cysteine (NAC). FACs-sorted ILC2s were cultured for 48 h with the vehicle, 5D, or the combination of 5D and NAC and were afterwards assessed for effector function. And indeed, proliferation and pro-inflammatory intracellular cytokine production were restored after addition of NAC in the culture (Fig. 5I–K). Altogether, our data demonstrate a previously unknown mechanism by which Ca²⁺ entry through Orai channels regulates pro-inflammation through modulation of cellular mitochondrial function.

**Orai inhibition downregulates ILC2-mediated airway hyperreactivity in vivo**
To explore the direct effect of Orai channels on ILC2 effector function in vivo, we first subjected BALB/c genetic background wild-type mice to the physiologically relevant setting utilizing *Alternaria alternata* (*A. alternata*), an allergen known to indirectly stimulate ILC2s[24] with the addition of intraperitoneal (i.p.) injections of either 5D (0.02 mg/mouse) or the vehicle (Fig. 6A).

Mice given the injections of 5D had lower levels of lung resistance and the development of airway hyperreactivity (Fig. 6B) as compared to those that received the vehicle. Furthermore, mice treated with 5D

harbored fewer eosinophils infiltrating the BAL (Fig. 6C). Consistent with these results, we found fewer lung ILC2s in mice treated with 5D, as well as a smaller percentage of ILC2s that produce IL-5, their primary proinflammatory cytokine responsible for eosinophil recruitment (Fig. 6D, E). Excitingly, we also saw a statistical decrease in the development of a chronic model of *A. alternata* murine lung inflammation, as measured by airway hyperreactivity, number of lung eosinophils in the BAL, as well as number of ILC2s in the lung (S4A-D). To further focus on the direct role of Orai channels in ILC2 effector function in vivo, we again utilized FACS-sorted pulmonary ILC2s from inducible *Orai1⁻/⁻* and *Orai1/2⁻/⁻* mice, as well as control ILC2s. The Orai-deleted and control ILC2s were adoptively transferred into cohorts of Rag2⁻/⁻ γc⁻/⁻ mice, mice that lack all B, T, and NK cells, including ILC2s. Mice were then subjected to the *A. alternata* protocol outlined above, and lung function was assessed on day 4 (Fig. 6F). As expected, mice adoptively transferred with *Orai1⁻/⁻* demonstrated lower AHR and fewer eosinophils than those given the control ILC2s (Fig. 6G, H). Mice adoptively transferred the knock-out cells also demonstrated fewer ILC2s found in the lung after day 4, though the same number of cells was transferred day 0 (Fig. 6I). Strikingly, *Orai1/2⁻/⁻* transferred mice had an even greater effect, suggesting that

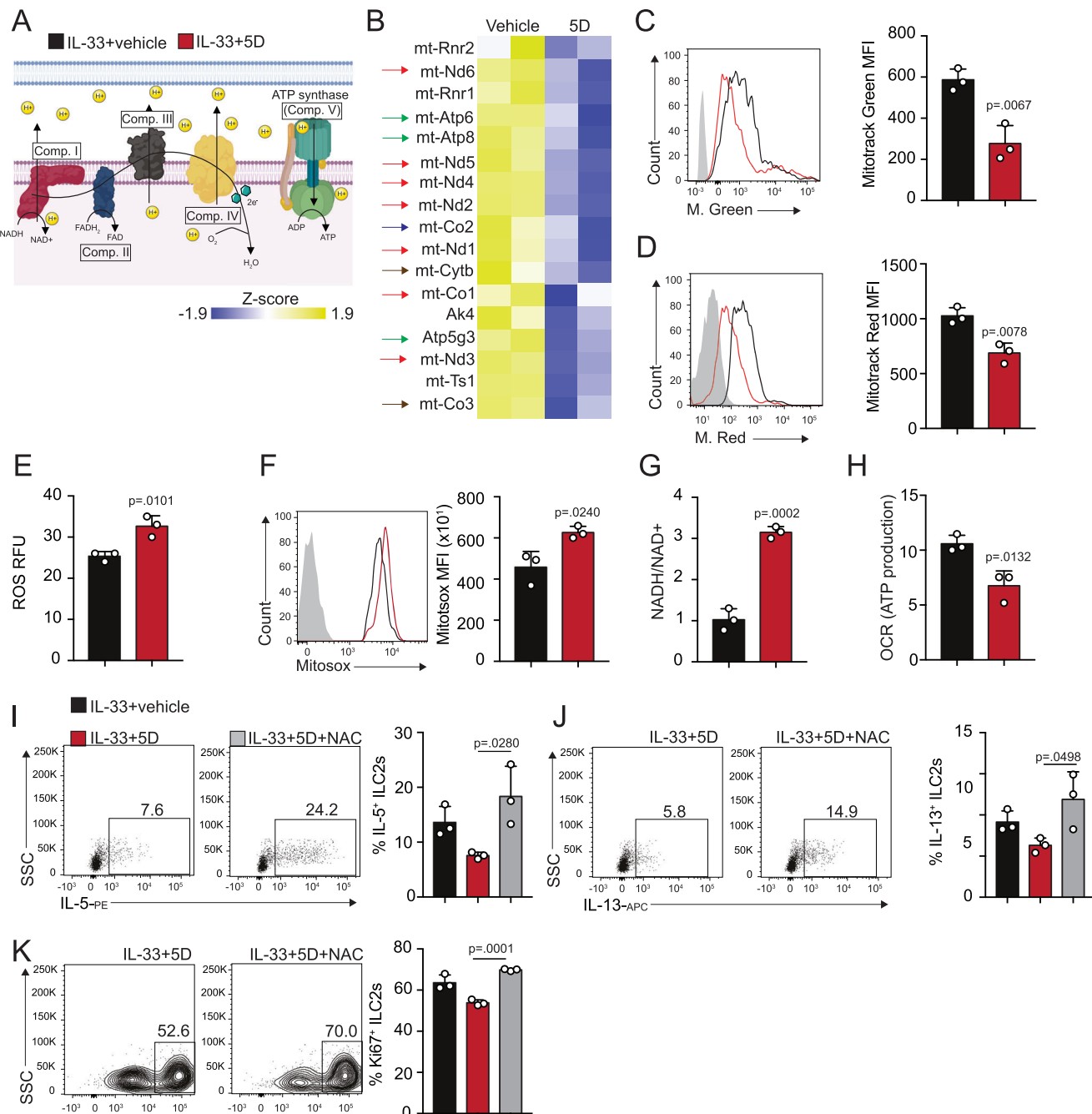

**Fig. 5 | Orai channels downregulate mitochondrial function, affecting ILC2 effector function and cytokine production.** **A** Schematic of the mitochondrial electron transport chain (mtETC) and the associated complexes. **B** Heatmap of statistically significant differentially expressed genes associated with the mtETC in ILC2s cultured with 5D or the vehicle. Arrow colors coordinate with the associated complex in (**A**). **C** Flow cytometry histogram and quantification of mitochondrial mass measured by Mitotracker Green in ILC2s cultured with 5D or the vehicle presented as mean ± SEM. $n$ = 3 biologically independent samples. **D** Flow cytometry histogram and quantification of mitochondrial membrane potential measured by Mitotracker Red in ILC2s cultured with 5D or the vehicle presented as mean ± SEM. $n$ = 3 biologically independent samples. **E** Total cellular reactive oxygen species (ROS) in treated and untreated ILC2s measured by microplate fluorescence presented as mean ± SEM. $n$ = 3 biologically independent samples. **F** Flow cytometry histogram and quantification of mitochondrial ROS in ILC2s cultured with 5D or the vehicle as measured by Mitosox presented as mean ± SEM. $n$ = 3 biologically independent samples. **G** NADH/NAD+ ratio quantification in ILC2s cultured with 5D or the vehicle measured by microplate reader presented as mean ± SEM. $n$ = 3 biologically independent samples. **H** ATP production presented as OCR measured by Seahorse Mitostress Test assay presented as mean ± SEM. $n$ = 3 biologically independent samples. **I** Representative flow cytometry plots and quantification of intracellular IL-5 production in ILC2s cultured with the vehicle, 5D, or 5D and the addition of antioxidant NAC presented as mean ± SEM. $n$ = 3 biologically independent samples. **J** Representative flow cytometry plots and quantification of intracellular IL-13 production in ILC2s cultured with the vehicle, 5D, or 5D and the addition of antioxidant NAC presented as mean ± SEM. $n$ = 3 biologically independent samples. **K** Representative flow cytometry plots and quantification of proliferation in ILC2s cultured with the vehicle, 5D, or 5D and the addition of antioxidant NAC presented as mean ± SEM. $n$ = 3 biologically independent samples. Data are representative of two independent experiments and are presented as means ± SEM. Source data are provided as a Source data file. A two-tailed Student's $t$ test for unpaired data was applied for comparisons between two groups except for multi-group comparisons where Tukey's multiple comparison one-way ANOVA tests were used. Figure 5A was created with BioRender.com.

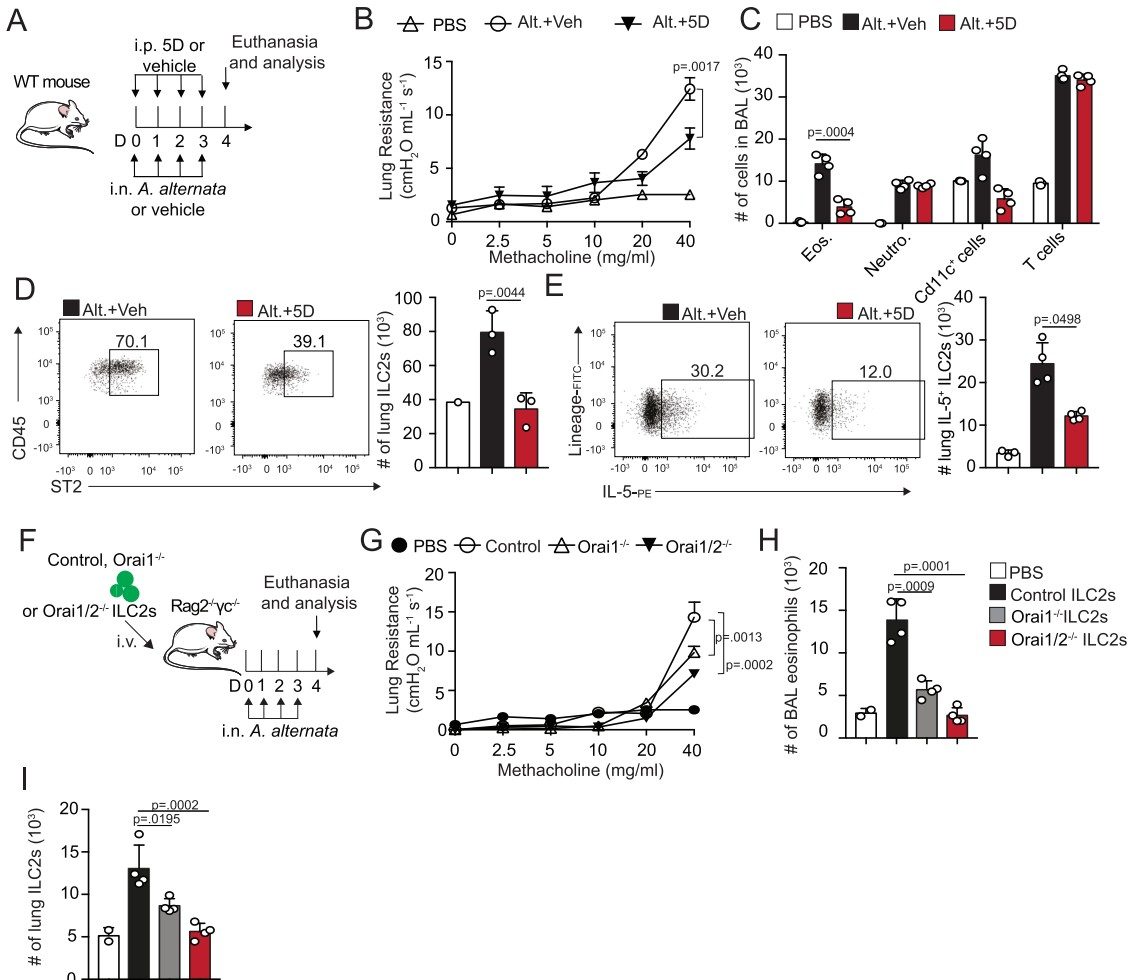

**Fig. 6 | Inhibition of Orai channels in pulmonary ILC2s downregulates development of airway inflammation. A** A cohort of BALB/cBYJ mice were challenged intranasally with *Alternaria alternata* (*A. alternata*) for four consecutive days. Mice were also given injections of 5D or vehicle. On day 4, AHR (**B**) was assessed. Additionally total number of eosinophils in the BAL (**C**), number of lung ILC2s (**D**), and IL-5 producing ILC2s (**E**) is presented as mean numbers +/− SEM. *n* = 4 biologically independent mice. **F** Pulmonary ILC2s from WT control, *Orai1*[−/−] and *Orai1/2*[−/−] mice were sorted into separate populations and deletion was induced as previously described. $5 \times 10^4$ ILC2s from the indicated populations were intravenously

injected to Rag2[−/−] γc[−/−] host mice. 24 h after transfer, mice were challenged intranasally with *Alternaria alternata* for four consecutive days. On day 5, AHR (**G**) was assessed. Additionally total number of eosinophils in the BAL (**H**) and total number of pulmonary ILC2s (**I**) is presented as mean numbers +/− SEM. *n* = 4 biologically independent mice. Data are representative of two independent experiments and are presented as means ± SEM. Source data are provided as a Source data file. Tukey's multiple comparison one-way ANOVA tests were used for multi-group comparisons. Mouse image designed with Servier Medical Art.

Orai2 has the ability to partially rescue the deletion of Orai1 in the context of ILC2 effector function, as also seen in Fig. 2F. Together, our data suggest that Orai1 and Orai2 support pathogenic ILC2 effector function in vivo, driving the development of ILC2-dependent airway hyperreactivity.

### Functional Orai channels enhance the induction of AHR in hILC2 recipient mice

We next investigated whether our results in murine models would translate to human ILC2s. Peripheral blood ILC2s from healthy donors were FACS-sorted and cultured with IL-33 to measure the expression of Orai channels (Fig. 7A). Human ILC2s do indeed express Orai1 and Orai2, both at the naive state and after IL-33 stimulation (Fig. 7B, C). The cells were then cultured with 5D or vehicle and the supernatant was collected to measure cytokines by ELISA.

As expected, inhibition of the Orai channels downregulated proinflammatory cytokine production in the human ILC2s (Fig. 7D). To investigate if this would translate therapeutically, we utilized our previously described humanized mouse model[25]. Human IL-5 has been shown to activate murine eosinophils emphasizing the feasibility of

using humanized mice in eosinophilic inflammatory studies[26,27]. Here, we isolated human peripheral ILC2s from healthy donors and cultured them with recombinant human (rh)IL-7 and rhIL-2 for 48 h (Fig. 7E). The cells were then adoptively transferred into Rag2[−/−] γc[−/−] mice and the mice were challenged for 3 consecutive days with rhIL-33 intranasally to induce an ILC2-dependent murine model of asthma. Mice were also given i.p. injections of 5D or the vehicle and lung inflammation was measured on day 3. Excitingly, mice given the Orai inhibitor demonstrated lower AHR development (Fig. 7F), as well as fewer eosinophils in the BAL and human ILC2s found in the lungs (Fig. 7G, H). Altogether our results using humanized mice suggests this therapeutic drug 5D targeting the Orai pathway has the potential to alleviate asthma symptoms in patients.

### Discussion

Overall this study introduces a unique mechanism by which $Ca^{2+}$ entry through Orai channels modulate proinflammatory ILC2 effector function and the development of airway hyperreactivity in a variety of pharmacological murine models of lung inflammation. We show that specific inhibition of Orai1 and Orai2 on pulmonary ILC2s leads to a

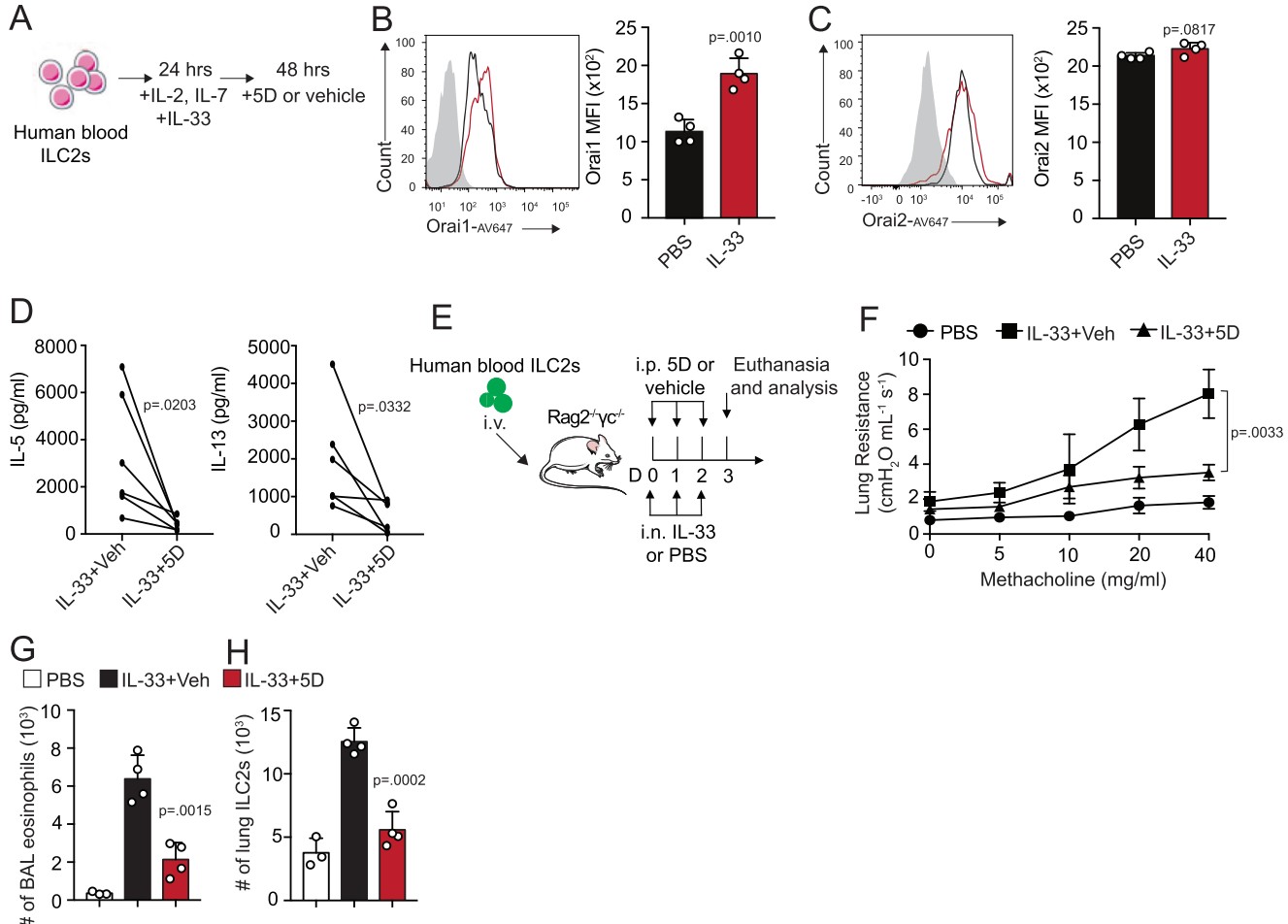

**Fig. 7 | Inhibition of Orai channels ameliorates human ILC2-mediated AHR.**
**A** FACS-sorted human blood ILC2s were cultured with rhIL-2, rhIL-7, and rhIL-33 for 24 h. 5D or vehicle were added to the culture for an additional 48 h. **B** Flow cytometry histogram and quantification of Orai1 expression after 24 h with survival cytokines in (**A**) presented as mean ± SEM. $n$ = 4 biologically independent samples. **C** Flow cytometry histogram and quantification of Orai2 expression after 24 h with survival cytokines in (**A**) presented as mean ± SEM. $n$ = 4 biologically independent samples. **D** After 48 h with 5D or vehicle, supernatant was collected and IL-5 and IL-13 levels were measured by ELISA. $n$ = 6 biologically independent samples. **E** FACS-sorted human blood ILC2s from healthy donors were adoptively transferred into Rag2$^{-/-}$ γc$^{-/-}$ mice. Mice were intranasally challenged with rhIL-33 (1 µg) or PBS and treated with i.p. injections of 5D or vehicle days 0, 1, and 2. Mice were euthanized on day 3 and AHR (**F**) was measured presented as mean ± SEM. $n$ = 4 biologically independent samples. **G** Total number of eosinophils infiltrating the BAL presented as mean ± SEM. $n$ = 4 biologically independent samples. **H** Total number of human ILC2s found in the lung presented as mean ± SEM. $n$ = 4 biologically independent samples. Data are representative of two independent experiments and are presented as means ± SEM. Source data are provided as a Source data file. A two-tailed Student's $t$ test for unpaired data was applied for comparisons between two groups, except for multi-group comparisons where Tukey's multiple comparison one-way ANOVA tests were used. Mouse and human ILC2 cell images designed with Servier Medical Art.

significant decline of functional metabolic pathways as well as the direct downregulation of mtETC proteins. Mechanistically, this causes an increase in mitochondrial ROS production and an uncoupling of the mitochondrial electron transport chain. Further, Orai inhibition severely downregulates ILC2 effector function and cytokine production in vivo and in vitro, potentially offering a therapeutic application of the CRAC channels in treating ILC2-dependent diseases.

To our knowledge, our group is the first to report the role of Orai channels and Ca$^{2+}$ entry in the health and homeostasis of pulmonary ILC2s, specifically in the context of AHR. Currently, therapeutic measures for asthma treatment involving calcium include the blockers for *chloride channels* (e.g., sodium cromoglicate and nedocromil sodium) and the *Ca$^{2+}$-activated K$^+$ channel, KCa3.1*[28–30]. KCa3.1 channels cause efflux of K$^+$ ions to maintain the driving force for Ca$^{2+}$ entry via the CRAC channels. These studies indicate that targeting the activation of CRAC channels would be excellent drug targets for asthma therapy. The widely used small molecule blockers of the Ca$^{2+}$-NFAT pathway including cyclosporine A or FK506 have a broad range of side effects because of their ubiquitous function[31,32]. Unlike calcineurin, CRAC

channels have been shown to play a primary role in immune cell types including T cells, B cells and mast cells, and thus Orai blockers are expected to be more specific with fewer side effects. Importantly, our humanized mouse model demonstrates the potential for 5D as a therapeutic application against ILC2-dependent lung inflammation. One limitation of the study is that injection of the drug in the humanized mouse model is not acting only on the transferred human ILC2s. However this model, that is independent of adaptive immunity, in combination with the *Orai1/2$^{-/-}$* adoptive transfer illustrate the exciting potential for the inhibition of Orai channels on ILC2s as a means of treatment in context of human AHR. The potency of compound 5D in the alleviation of allergic asthma suggests an alternative mechanism to the conventionally-used long lasting beta-2 agonists in the treatment of asthma. Beta-2 agonists work on relaxing the smooth muscles of the bronchi, targeting a symptom of the pathophysiology of the disease. Here, we suggest a method that targets the initial onset of the disease at a cell-based level. Further studies will have to be conducted to compare the efficacy of this method to beta-2 agonists. Although recent years have revealed that ILC2s play an important role in a broad

range of immune-mediated disorders, a major gap exists between understanding ILC2 biology and modulating these cells for therapeutic application.

We observed in our study that pulmonary mouse and human ILC2s express both Orai1 and Orai2 channels after IL-33 stimulation. Interestingly, while both isoforms are expressed at basal levels in naive ILC2s, Orai1 appears to be highly inducible after activation, while Orai2 is only slightly enhanced in both murine and human ILC2s. Grimes et al. recently detailed Orai isoform expression on neutrophils and their effect on neutrophil activation[33]. They also found basal levels of Orai1 and Orai2 on resting neutrophils, expressed at a 1:1 ratio, while activation by inflammatory agents raised the ratio to 30:1[33]. They concluded the ratio change was mainly due to the elevation of Orai1 expression after exposure[33]. We believe a similar phenomenon is occurring in activated ILC2s. The signaling pathways regulating Orai1 and Orai2 induction by IL-33 stimuulation in ILC2s is currently unknown. However, previous reports demonstrate that IL-33 upregulates STIM1 via p38 and AP-1 signaling in epithelial cells[34]. Whether this is true for ILC2s requires further invesitagion. Inhibition of both Orai1 and Orai2 through therapeutic 5D or genetically engineered double knock-out mice leads to a significant downregulation in proliferation and cytokine production, including IL-5, IL-13, GM-CSF, IL-9 and IL-6. The role of Orai1 in the regulation of immune effector function has been well established, but investigation into the alternative isoforms and their role is lacking. Recently Vaeth et al. demonstrated that Orai1 deletion in T cells reduces effector function, while specific deletion of Orai2 enhances SOCE in both T cells and macrophages[12]. Alternatively, in neutrophils, deletion of Orai2 was shown to reduce SOCE and negatively affect effector function[33]. The effect of genetically deleting only Orai2 on ILC2s requires further investigation. Regardless, both studies found that double deletion of Orai1 and Orai2 on T cells or neutrophils reduces the cells' ability to pathogenetically function in their respective disease setting[12,33]. Similarly in such settings, we discovered that blocking Orai channels specifically in pulmonary ILC2s in vivo efficiently ameliorated the development of ILC2 dependent AHR stimulated by *A. alternata* or IL-33 in both murine models and humanized mouse models of lung inflammation. We observed lower levels of AHR, as well as fewer infiltrating BAL eosinophils and IL-5-producing ILC2s in the lungs. Our results suggest Orai channels play a crucial role in the development of lung inflammation in ILC2-dependent models of asthma.

Our group here has demonstrated that Orai channel inhibition severely alters mitochondrial health and function, potentially offering a mechanism by which $Ca^{2+}$ concentration controls pathogenic immune function. Previously it has been shown in CD4 + Th17 cells that inhibition of CRAC channels severely affected mitochondrial homeostasis, resulting in similar observations of increased ROS production as well as reduced immune cell effector function[18]. $Ca^{2+}$ entry is well known to assist mitochondria in the production of ATP through the increase in production and consumption of NADH involved in Complex I and V in the mtETC[35]. Traditionally, mitochondrial membrane potential and mass can be indicative of effective mtETC function and ATP synthesis. Additionally, lower mitochondrial membrane potential classically correlates with lower levels of ROS production[36]. In our case however we see a decrease in membrane potential in treated cells correlated with an increase in mtROS production. We suspect the cells with inhibited Orai channels are undergoing mtETC uncoupling. In these cases, mitochondrial ETC uncoupling, especially in the impairment of ADP-coupled oxidative phosphorylation, will lead to an increase in ROS production primarily due to a defect in the cellular antioxidant system[23], supported by our data demonstrating supplementation of antioxidants correlates with increased ILC2 proliferation and cytokine production. The ETC uncoupling is confirmed by the increased NADH/NAD+ ratio in conjunction with the decreased ATP production in cells

treated with the inhibitor, as mtETC uncoupling leads to the cell focusing on the generation of ROS rather than the maintenance of ATP production[37]. What remains unclear however is where specifically in the pathway the inhibition of $Ca^{2+}$ flux is acting to induce such uncoupling. One option is that the inhibition of the $Ca^{2+}$ flux itself results in defects in the antioxidant system, subsequently leading to an uncontrolled rise in ROS production. Excessive ROS has been linked to mitochondrial DNA damage, similar to that seen in our RNA-seq data, and defects in the complexes can lead to an uncoupling of the mtETC[38]. An alternative option is that the Orai inhibition is acting directly to downregulate the mtETC genes, resulting in defects in the chain. Defects could lead to a lower membrane potential as well as the uncoupling of the ETC, resulting to higher ROS levels that overwhelm the antioxidant system. Regardless, excessive ROS has been well linked to insufficient ATP production, impairment of the mitochondria, DNA damage and eventually cell death[39]. Though the details of how mtROS interferes with gene transcription is still being investigated, mitochondrial (mt)DNA and mitochondrial reactive oxygen species (mtROS) have been shown to be linked to cascades of activation and inhibition transcription factors, including the CRAC-associated NFAT and inflammasome NLRP3, providing further evidence that metabolism and immune responses are intimately dependent on one another[21].

In summary, in this report we reveal a previously unrecognized immunoregulatory role for inhibition of Orai channels on ILC2s in the context of murine and humanized mouse AHR development. Our results suggest a crucial role for Orai channels in the functional mitochondrial electron transport chain and subsequent proinflammatory effector function. Further investigation into the signaling pathways of how this defect affects cytokine production are warranted, as mitochondrial control of nuclear genes is largely unknown. However, findings from this study could lead to identification of therapeutic targets that can be pharmacologically manipulated to specifically modulate ILC2s for the treatment of asthma. More broadly, this study provides a unique paradigm for metabolically manipulating other complex cells as well. ILC2s are well established to be present in mucosal tissue and their overproduction of proinflammatory cytokines has been linked to a significant variety of autoimmune disease and disorders[4]. The results of our studies will therefore have potential implications in other diseases involving ILC2s, including allergic diseases such as atopic dermatitis, allergic rhinitis and inflammatory bowel disease.

## Methods
### Mouse experiments
Experimental protocols were approved by the USC institutional Animal Care and Use Committee (IACUC) and conducted in accordance with the USC Department of Animal Resources' guidelines. Age (5–8 weeks) and sex matched mice were used in the studies. BALB/cByJ, and $Rag2^{-/-}$ $γc^{-/-}$(C;129S4-Rag2tm1.1Flv Il2rgtm1.1Flv/J) mice on a BALB/c background were bred in the animal facility at the Keck School of Medicine, University of Southern California (USC), and maintained at a macroenvironmental temperature of 21–21–22 °C, humidity (48–52%), in a conventional 12:12 light/dark cycle with lights on at 6:00 a.m. and off at 6:00 p.m. Orai1-deficient mice have been previously described[40]. Orai2-deficient mice were generated using CRISPR/Cas9-mediated recombination. Four sgRNAs targeting exon 2 (the first coding exon) of the mouse *Orai2* gene were designed and examined for recombination in situ, of which two sgRNAs were chosen for injection (PNABio Inc.). The following sgRNA sequences were used: sgRNA1: 5' *gccgggttcagg gcaggcagggg* 3' and sgRNA2: 5' *gggcatggattaccgagactggg* 3'. Purified sgRNAs together with Cas9 mRNA were injected into day 0.5 single-cell embryos from $Orai1^{fl/fl}$ mice and transferred into the oviducts of pseudopregnant recipient mice (Genome modification facility, University of Southern California). The resulting pups were screened by PCR to identify those that had successful deletion of 43 nucleotides

between the two sgRNA sequences and validated by sequencing (Figure S1). The mice with desired deletion were bred to Tg(*UBC-Cre/ERT2*) mice (The Jackson Laboratory stock no. 007001) to generate *Orai1*$^{fl/fl}$; *Orai2*$^{-/-}$; *UBC-Cre/ERT2* mice.

## Flow cytometry

The following murine antibodies were used: anti-mouse FITC CD3e (145-2C11), CD5 (53-7.3), TCRβ (H57-597), CD45R (RA3-6B2), Gr-1 (RB6-8C5), CD11c (N418), CD11b (M1/70), Ter119 (TER-119), FcεRIα (MAR-1), TCRγδ (eBioGL3), CD335 (29A1.4), PE-Cy7 anti-mouse CD127 (A7R34), APCCy7 anti-mouse CD45 (30-F11), PECy7 anti-mouse CD45 (30-F11), APCCy7 anti-mouse CD11c (N418), were purchased from BioLegend. PE anti-mouse SiglecF (E50-2440) was purchased from BD Biosciences. PerCP-eFluor710 anti-mouse ST2 (RMST2-2), eFluor450 anti-mouse CD11b (M1/70) were purchased from ThermoFisher. Rabbit anti-mouse anti-human Orai1 and Orai2 antibodies were purchased from Novus Biologicals. Secondary antibody Alexa Fluor 647 goat anti-rabbit wass purchased from Jackson ImmunoResearch Laboratories. Intranuclear staining was performed using the Foxp3 Transcription Factor Staining Kit (Thermofisher) per the manufacturer's instructions. APC anti-mouse Ki67 (SolA15, Thermofisher) was used. Intracellular staining was performed using the BD Biosciences Cytofix/Cytoperm kit. When indicated, cells were stimulated in vitro for 4 h with 50 μg/mL PMA, 500 μg/mL ionomycin (both Sigma) and 1 μg/mL Golgi plug (BD Biosciences) before cytokine assessment. APC anti-mouse IL-13 (85BRD, Thermofisher), PE anti-mouse IL-5 (TRFK5, BioLegend) were used. All antibodies listed above were used at a 1:300 dilution, except the intracellular antibodies at 1:100. Live/dead fixable violet cell stain kit was used to exclude dead cells (Thermofisher) and CountBright absolute counting beads (Thermofisher) to calculate absolute cell numbers when indicated. For analysis of the mitochondria, cells were stained with 40 nM MitoTracker Green or Red (Life Technologies) for 20 min or MitoSox Red (Life Technologies) for 10 min, respectively, at 37 °C. For glucose uptake measurements, cells were incubated in media containing 50 μg/ml 2-NBDG (Thermo Fisher Scientific) for 20 min at 37 °C after surface antibody staining. To measure lipid droplet quantification, cells were incubated in media containing 1000 ng/ml Bodipy or Bodipy FL C16 (Thermo Fisher Scientific) at 37 °C for 30 min. Stained cells were analyzed on FACSCanto II and/or FACSARIA III systems and the data was analyzed with FlowJo version 10 software.

## Murine ILC2 and in vitro culture

Murine ILC2s were FACS-sorted to a purity of >95% on a FACSARIA III system. Isolated ILC2s were cultured at 37 °C (5–7 × 10⁴/mL) for 24 h as indicated with N-{2,2,2-trichloro-1-[(naphthalen-2-yl)amino]ethyl} furan-2-carboxamide (5D solubilized in DMSO, purchased from Asinex) and rmIL-2 (10 ng/mL), rmIL-7 (10 ng/mL) survival cytokines. After 24 h, +/−rmIL-33 (10 ng/ml) was added for an additional 48 h in complete RPMI (cRPMI). For cRPMI, RPMI (Gibco) was supplemented with 10% heat-inactivated FBS (Omega Scientific), 100 units/mL penicillin and 100 mg/mL streptomycin (GenClone). In the indicated experiments, N-Acetyl-L-cysteine (NAC, 10 μM, Sigma) was included in the culture.

## Human ILC2 isolation and in vitro culture

Experimental protocols were approved by the USC Institutional Review Board (IRB) and conducted in accordance with the principles of the Declaration of Helsinki. Human blood ILC2s were isolated from total peripheral blood mononuclear cells (PBMCs) to a purity of >95% on a FACSARIA III system. A total of 14 healthy volunteers (7 males, 7 females) with written consent participated in the study, aged 18–70, no compensation was provided. Briefly, human fresh blood was first diluted 1:1 in PBS 1X and transferred to SepMate™-50 separation

tubes (STEMCELL Technologies) filled with 12 mL Lymphoprep™. Samples were centrifuged for 10 min and PBMCs were collected. CRTH2⁺ cells were then isolated using the CRTH2 MicroBead Kit, used according to the manufacturer's conditions. Samples were then stained and ILC2s were isolated based on the absence of common lineage markers (CD3, CD5, CD14, CD16, CD19, CD20, CD56, CD235a, CD1a, CD123), and the expression of CD45, CRTH2 and CD127. Isolated ILC2s were cultured at 37 °C (5 × 10⁴/mL) with recombinant human (rh) IL-2 (10 ng/mL) and rhIL-7 (10 ng/mL) in cRPMi.

## Culture measurements

Murine or human ILC2s were FACS-sorted and cultured in media containing survival cytokines and 5D when indicated for 24–48 h. The levels of cytokines present in culture supernatants were measured by customized 7-panel mouse Legendplex kit or human cytokine ELISA kits according to the manufacturer's instructions (BioLegend). Supernatants were analyzed for L-lactate levels using the Glycolysis Cell-Based Assay Kit (Cayman Chemicals). Total cellular ROS was measured utilizing the Cellular Ros Assay kit (Red) from Abcam. NAD/NADH was measured by NAD/NADH Assay Kit (Colorimetric) also from Abcam according to the manufacturer's protocol.

## Metabolic flux analysis

The real-time extracellular acidification rate and oxygen consumption rate (OCR) were measured with a Seahorse XF HS (Higher sensitivity) analyzer (Seahorse Bioscience, North Billerica, Mass). Briefly, 50,000 activated lung ILC2s were plated in Seahorse medium supplemented with 1 mmol/L pyruvate, 2 mmol/L glutamine, and 10 mmol/L glucose. The Mito Stress Test Kit (Agilent Technologies, San Diego Calif) with 1 mmol/L oligomycin, 4 mmol/L carbonyl cyanide-4-(trifluoromethoxy) phenylhydrazone (FCCP), and 0.5 mmol/L rotenone (Rot) and antimycin A was used, according to the manufacturer's protocol. The Glycolysis Rate Assay with 0.5 μM rotenone (Rot) and antimycin A and 50 mM 2-deoxy-D-glucose was used, according to the manufacturer's protocol.

## Calcium flux analysis

WT ILC2s were FACS-sorted and cultured in media containing survival cytokines for 24 h. Cells were then loaded with 100 μg/mL Fluo-8 AM for 30 min in calcium-free HBSS buffer. Cells were then pretreated for 15 min with 1 mM thapsigargin to passively deplete Ca²⁺ stores. SOCE was measured by exchanging the Ca²⁺-free HBSS with that containing 2 mM CaCl₂, with the addition of 5D or the vehicle in the solution. Fluorescence was measured by flow cytometry.

## In vivo experiments and tissue preparation

For induction of lung inflammation, when indicated mice were challenged for 4 consecutive days with 100 μg *Alternaria alternata* extracts i.n. in 50 μl or PBS. A cohort was also given i.p. injections of 5D or vehicle. On day 5, AHR was measured and BAL was collected, as described below. The lungs were perfused with PBS and digested in Collagenase IV (400 U/mL, MP Biomedicals, LLC) for 1 h at 37 °C[41]. The lungs were then stained with antibodies to identify ILC2s. ILC2s were gated as lineage (CD3e, CD5, CD45R, Gr-1, CD11c, CD11b, Ter119, TCRγδ, TCRβ, CD335 and FCεRIα) negative, CD45⁺, ST2⁺, CD127⁺ cells. For the chronic model, mice were challenged for 3 consecutive days with 5 μg *Alternaria alternata* extracts i.n. in 50 μl. Mice were then challenged every Monday and Friday for the following three weeks. A cohort was also given i.p. injections of 5D or vehicle. On 26, AHR, BAL eosinophils and lung ILC2s were assessed.

## Adoptive transfer

Indicated populations of mouse or human ILC2s were isolated as above and cultured for 24 h with survival cytokines[42]. 5.0 × 10⁴ of the appropriate population of cells were transferred in PBS to Rag$^{-/-}$γc$^{-/-}$ mice by tail intravenous (i.v.). 24 h later, 0.5 μg rmIL-33 i.n. in 50 μL was given

once a day for 3 days. A cohort was also given i.p. injections of 5D or vehicle. AHR was then measured on day 4. Alternatively, 100 µg *Alternaria Alternata* extracts i.n. in 50 µl was given once a day for 4 days. AHR was then measured on day 5.

## Measure of airway hyperreactivity

Lung function was evaluated by direct measurement of lung resistance using the FinePointe RC system (Buxco Research Systems, Wilmington, NC) under general ketamine and xylazine anesthesia[43]. AHR was measured by exposure to an aerosol containing increasing doses of Methacholine (Sigma), following a baseline measurement after the delivery of a PBS aerosol. Maximum lung resistance values were recorded during a 3-min period after each methacholine challenge.

## BAL collection

The trachea was cannulated, the lungs lavaged three times with 1.0 ml PBS and the collected fluid pooled[44]. Eosinophils were gated as CD45+ CD11c- SiglecF+ single cells.

## RNA sequencing and data analysis

For each sample, a total of 10 pg of RNA was used to generate cDNA (SMARTer Ultra Low Input RNA v3 kit, Clontech) for library preparation. Samples were then amplified and sequenced on a NextSeq 500 system (Illumina) where on average 30 million reads were generated from each sample. Raw reads were then further processed on Partek Genomics Suite software, version 7.0; Partek Inc. Briefly, raw reads were aligned by STAR – 2.6.1d with mouse reference index mm10 and GENECODE M21 annotations. Aligned reads were further quantified and normalized using the upper quartile method and differential analysis by Gene Set Analysis (GSA). Transcripts showing an average normalized count below 1 were removed from the analysis, as were genes showing cumulative normalized counts below 10. Pathway analysis was performed by using Qiagen Ingenuity Pathway Analysis software. Single-cell RNA sequencing data was downloaded and reanalyzed as described previously[45].

## Statistical analysis and figure design

Data presented in the Figures are from a single (non-pooled) representative experiment, with the appropriate sample size and number of repeats indicated in the legends. A two-tailed Student's *t* test for unpaired data was applied for comparisons between two groups, except for multi-group comparisons where one-way analysis of variance tests were used. All tests were performed using Prism Software (GraphPad Software Inc.). ILC2 cell and mouse images on Figs. 2, 6 and 7 were designed with Servier Medical Art (https://smart.servier.com). Figure 5A was created with BioRender.com.

## Reporting summary

Further information on research design is available in the Nature Portfolio Reporting Summary linked to this article.

## Data availability

The RNA-seq data from Fig. 3 have been deposited in the Genbank database under the GEO accession code GSE221009. Single-cell RNA sequencing data is found in the Genbank database under GEO accession code GSE102299. All data are included in the Supplemental Information. The raw numbers for charts and graphs are available in the Source data file whenever possible. Source data are provided with this paper.

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

## Acknowledgements

This article was financially supported by National Institutes of Health Public Health Service grants R01HL144790, R01 HL151493, R01 HL159804 and R01 AI145813 (O.A.); R01AI146615, R21AI149236 (Y.G.), R01AI146352, and R21 AI30653 (S.S.).

## Author contributions

E.H. performed, analyzed experiments and wrote the first draft of manuscript under O.A. supervision. B.H., D.G.H., P.S.-J., and J.P. performed experiments and maintained animal husbandry. S.H., S.S., and Y.G. provided valuable reagents and performed experiments related to genetically modified animal models. O.A. supervised, designed the experiments, interpreted the data and finalized the manuscript. All authors critically read the manuscript.

## Competing interests

The authors declare no competing interests.
