## [Peer Review File · Nature Communications]

Orai inhibition modulates pulmonary ILC2 metabolism and alleviates airway hyperreactivity in a mouse modelREVIEWER COMMENTS

Reviewer #1 (Calcium signaling) (Remarks to the Author):

Group 2 innate lymphoid cells (ILCs) play a central role in driving immune responses, particularly in regions exposed to the external environment such as the airways and GI tract. Here, these sentinel cells respond to various alarmins released from epithelia and then secrete a battery of pro-inflammatory cytokines and chemokines that play an important role in orchestrating the subsequent immune response. However, despite their significance and unexplored clinical relevance, little is known about how these cells are activated and what their role is in specific diseases. The new study by Srikanth, Gwack, Akbari and colleagues breaks new ground by establishing how ILCs are activated and the underlying molecular mechanism which involves the Orai1 channel and altered mitochondrial metabolism. The authors further show that ILCs play a particularly prominent role in airway hyperreactivity, a feature of many airways diseases including asthma, COPD and COVID19-triggered respiratory distress. Finally, the authors demonstrate that an Orai channel blocker has significant therapeutic benefit.

This is an elegant study, using powerful state of the art approaches, that tackles an important but overlooked area and which convincingly demonstrates a central role of ILCs in airway disease. The work is carefully designed, easy to read, the data are of high quality and the conclusions are persuasive. In my opinion, the paper is acceptable for publication in Nature Comms with only very minor revisions.

1. The authors use compound D to show that block of CRAC channels leads to rescue of metabolic phenotypic switch in ILCs. Is compound D selective for Orai1? And the authors incubate cells in the drug for 48-72 hours. Are there any cytotoxic effects and is store calcium content normal after such exposure? This could be easily measured through the ionomycin Ca^{2+} transient. And does the IC_{50} for CRAC channels change when cells are exposed to compound D for 48 hours? .
2. The effects of compound D on airway hyperreactivity are dramatic and lead to an almost complete reversal. These are exciting findings of major translational relevance. In light of this, it might be worthwhile comparing compound D with, a long lasting β_2 agonist for example, to show the impressive efficacy of the compound.
3. In the mouse model of hyperreactivity, the authors comment on the reduced number of ILCs in the presence of compound D. This is very interesting. I wonder whether the secretion from the remaining ILCs is reduced. And for the numbers to fall, is there less recruitment or reduced proliferation?

These are all minor comments and could be addressed through changes to the text. Overall, this is an excellent paper.

Reviewer #2 (Lung inflammation, Orai/ Ca^{2+} signaling) (Remarks to the Author):

This study investigated the function of Orai1 and Orai2 on human and murine ILC2s. The authors demonstrate expression of both Orai1 and Orai2 on human and murine ILC2s. Inhibition of the channels results in reduced airway hyperreactivity (AHR). Blocking Orai channels downregulates fatty acid oxidation/Oxphos and glycolysis in proinflammatory ILC2s. Mitochondrial electron transport chain is downregulated, and ROS is upregulated. These results may open new therapeutic targets in the treatment of allergic asthma. These studies are well done, demonstrating the significant role of Orai1 and Orai2 in ILC2s. The authors have used a variety of approaches and techniques to elucidate these roles.

Combination of in vitro and in vivo studies in addition to using a humanized mouse model further demonstrate the relevance of their findings.

Major Comments:

1. Although interesting, the studies investigating mitochondria and metabolism do not necessarily fit the overall focus of the manuscript.
2. For the in vivo studies, why was the *Alternaria* treatment only done for 3 days? Why did the authors choose to start the 5D treatment at the same time as the *Alternaria* treatment?
3. Why was the time schedule for the humanized mouse studies (2 days of treatment, Figure 7E) different from the *Alternaria* studies?
4. How do the authors explain the lesser response in lung resistance in Figure 7E in comparison to both Figure 6B and G?

Reviewer #3 (ILC, type 2 immunity) (Remarks to the Author):

General comments:

In this manuscript, the authors examined the roles of Orai channels, the pore components of CRAC channels, in effector functions of ILC2s. ILC2s play key roles in type 2 immunity. However, our knowledge is relatively limited regarding the mechanisms to control ILC2s. No or few studies have been performed previously to dissect the roles of Ca²⁺ signaling pathway in ILC2s. This manuscript addresses this major gap in our knowledge, and therefore, it is novel and potentially important. The experiments are straightforward by using an Orai inhibitor and ILC2 that are genetically deficient in Orai and in vitro culture and in vivo adoptive transfer models. The authors also analyzed metabolic and mitochondrial homeostasis of ILC2s to dissect the mechanisms.

The authors concluded that Orai channels play a crucial role in the functional mitochondria electron transport chain and subsequent pro-inflammatory effector function of ILC2s. This conclusion is largely supported by the experimental data. However, there are several questions remaining regarding the physiologic functions of Orai in ILC2s and specificity of the Orai inhibitor. In addition, in vivo experiments would be more convincing if a model relevant to asthma was used.

Specific comments:

1. To the best of this reviewer's knowledge, this manuscript is the first to report the expression and function of Orai channels in ILC2s. While their expression is clearly demonstrated in Figure 1, the data to support the function of Orai are rather weak as it depends on the use of a pharmacological agent thapsigargin (Figure 2). Several fundamental questions remain regarding Orai channels in ILC2s. Are Orai channels involved in more physiological events, such as oscillation of Ca²⁺ at a resting state or stimulus-dependent Ca²⁺ influx, such as that induced by leukotrienes or potentially cytokines?
2. Many of the experiments in this manuscript dependent on a pan-Orai inhibitor 5D. The scientific rigor of this manuscript would be enhanced with more characterization of the effects of 5D. For example, what is the dose and effect relationship between 5D and ILC2 effector function? How were the doses of 5D used for in vitro and in vivo experiments selected? Is ILC2 apoptosis or viability affected when ILC2s are cultured with 5D for 72 hours as was done in Figure 2B through 2E? These specificity questions would be particularly important as Orai channel inhibition may alter mitochondrial health (Reference 18 of the manuscript).

3. In Figure 2F and 2G, Orai-deficient ILC2s were expanded by IL-33 and then type 2 cytokine expression was analyzed. Does Orai-deficiency affect proliferation and viability of ILC2s?

4. In Figure 2G, IL-5 production was minimally affected by the deficiency in Orai1 while IL-13 production was significantly inhibited. How can the authors explain the dissociation between these type 2 cytokines?

5. In Figure 2, the authors concluded that “blocking Ca²⁺ entry specifically through Orai channels paly a crucial and previously underrecognized role the effector function of ILC2s”. This statement appears to be rather an over-interpretation as the authors did not study IL-33-induced Ca²⁺ entry through Orai.

6. Similarly in Figure 4, the authors concluded that “calcium flux is indispensable for metabolism”. This statement also appears to be over-interpretation as the calcium flux was not examined. Can the calcium flux be demonstrated in the model used in Figure 4, and does 5D inhibit the flux? Does chelation of extracellular calcium or zero calcium medium (and therefore no calcium flux) have similar effects as shown in Figure 4?

7. In Figure 6, The role of ILC2s was examined in vivo by acute exposure to *Alternaria*. Because the authors propose the potential therapeutic application of 5D to treat asthma, a model more relevant to asthma could be used. Can ILC2s be suppressed by 5D during chronic airway inflammation?

Minor comments:

1. In Figure 6B, the effects of 5D on *Alternaria*-induced type 2 cytokine production and airway eosinophilia were examined. Does 5D affect migration of eosinophils?

2. In Figure 6D, all the lineage-negative cells appear to be inhibited by treatment with 5D. The gating strategy for lineage-negative cells and ILC2s need to be provided.

We thank the Reviewers for their constructive comments. We specially believe the Reviewers' criticism are reasonable and the additional data requested further strengthens our manuscript. We have performed numerous additional experiments and **addressed all concerns raised by the Reviewers**. These include experiments assessing further the in vitro characterization of 5D and Orai KO ILC2s, as well as the effect of the 5D on chronic models of murine asthma. Please see the detailed point by point response below, referencing the relevant sections of the manuscript. In the marked copy, these modifications have been highlighted in **green**.

Reviewer #1

Group 2 innate lymphoid cells (ILCs) play a central role in driving immune responses, particularly in regions exposed to the external environment such as the airways and GI tract. Here, these sentinel cells respond to various alarmins released from epithelia and then secrete a battery of pro-inflammatory cytokines and chemokines that play an important role in orchestrating the subsequent immune response. However, despite their significance and unexplored clinical relevance, little is known about how these cells are activated and what their role is in specific diseases. The new study by Srikanth, Gwack, Akbari and colleagues breaks new ground by establishing how ILCs are activated and the underlying molecular mechanism which involves the Orai1 channel and altered mitochondrial metabolism. The authors further show that ILCs play a particularly prominent role in airway hyperreactivity, a feature of many airways diseases including asthma, COPD and COVID19-triggered respiratory distress. Finally, the authors demonstrate that an Orai channel blocker has significant therapeutic benefit.

This is an elegant study, using powerful state of the art approaches, that tackles an important but overlooked area and which convincingly demonstrates a central role of ILCs in airway disease. The work is carefully designed, easy to read, the data are of high quality and the conclusions are persuasive. In my opinion, the paper is acceptable for publication in Nature Comms with only very minor revisions.

We thank the Reviewer for the constructive comments, please find below the response to the specific comments:

1. The authors use compound D to show that block of CRAC channels leads to rescue of metabolic phenotypic switch in ILCs. Is compound D selective for Orai1? And the authors incubate cells in the drug for 48-72 hours. Are there any cytotoxic effects and is store calcium content normal after such exposure? This could be easily measured through the ionomycin Ca²⁺ transient. And does the IC₅₀ for CRAC channels change when cells are exposed to compound D for 48 hours?

We apologize for the confusion. Compound 5D is selective for all the Orai channels, including Orai1, Orai2, and Orai3. We have now added a sentence in the Results section better introducing the inhibitor and clarifying this point. The Reviewer is correct, cytotoxicity is an important factor in the potential use of this drug for therapeutic purposes. We previously titrated the compound 5D concentration prior to our experiments, and we now include this data in the supplementary figure (**Figure S2A**). Similar to our experiments outlined in **Figure 2B**, we FACS-sorted pulmonary ILC2s and cultured with increasing concentrations of compound 5D or the vehicle for 24 hours. We then added IL-33 or PBS for an additional 48 hours. Viability was assessed by Annexin V and DAPI staining. We determined 5 μ M to be the most effective dose of compound 5D with no statistical effect on cellular viability. We believe this new data demonstrate the potential use of 5D for therapeutic purposes. Previously, we have reported the IC₅₀ of 5D (PMID: 24307733). While we do not have the IC₅₀ after 48 hours, we have used this compound extensively in cultures extending up to 4 days, while carefully assessing apoptosis

(PMID: 24307733). We therefore do not anticipate a major shift in IC50 at 48 hours. We have now provided the careful titration of the optimum 5D dose and have used the minimum effective dose in all our experiments. Further, it has previously been reported that complete loss of expression of either Orai1 alone, or both Orai1 and Orai2 does not affect ER store content in T cells (PMID 18591248 and 28294127), hence it is unlikely that compound 5D treatment will alter ER Ca²⁺ stores.

2. The effects of compound D on airway hyperreactivity are dramatic and lead to an almost complete reversal. These are exciting findings of major translational relevance. In light of this, it might be worthwhile comparing compound D with, a long lasting b2 agonist for example, to show the impressive efficacy of the compound.

We agree with the Reviewer that these results are very exciting and that comparisons with a long last b2 agonist would be very interesting. Although we believe this comparison is currently outside the scope of this specific publication, it has the potential to be an exciting separate manuscript focusing on the mechanism by which b2 agonists inhibit ILC2 proliferation and effector function. We've now included in the discussion an examination of this comparison as a promising future direction.

3. In the mouse model of hyperreactivity, the authors comment on the reduced number of ILCs in the presence of compound D. This is very interesting. I wonder whether the secretion from the remaining ILCs is reduced. And for the numbers to fall, is there less recruitment or reduced proliferation?

This is a very interesting point. Since we first injected the mice with the compound or vehicle 24 hours after the adoptive transfer of the pulmonary ILC2s, we have no reason to suspect there would be a difference in recruitment of the ILC2s to the lung based on the experimental group. Therefore, we believe it is reasonable that the difference in number is based on the decreased rate of proliferation. Further, we do believe that it is possible the secretion from the remaining cytokines is decreased when treated with 5D. While we only performed intracellular staining in this specific experiment, in vitro characterization of the treated ILC2s indicated that the concentration of secreted IL-5 and IL-13 is significantly decreased as compared to those treated with the vehicle.

These are all minor comments and could be addressed through changes to the text. Overall, this is an excellent paper.

Reviewer #2

This study investigated the function of Orai1 and Orai2 on human and murine ILC2s. The authors demonstrate expression of both Orai1 and Orai2 on human and murine ILC2s. Inhibition of the channels results in reduced airway hyperreactivity (AHR). Blocking Orai channels downregulates fatty acid oxidation/Oxphos and glycolysis in proinflammatory ILC2s. Mitochondrial electron transport chain is downregulated, and ROS is upregulated. These results may open new therapeutic targets in the treatment of allergic asthma.

These studies are well done, demonstrating the significant role of Orai1 and Orai2 in ILC2s. The authors have used a variety of approaches and techniques to elucidate these roles. Combination of in vitro and in vivo studies in addition to using a humanized mouse model further demonstrate the relevance of their findings.

We thank the Reviewer for their comments and have performed several experiments that we believe address the concerns raised. Please find our response below.

Major Comments:

1. Although interesting, the studies investigating mitochondria and metabolism do not necessarily fit the overall focus of the manuscript.

We thank the reviewer for their constructive criticism. We agree that the conclusions drawn from the mitochondria and metabolism assessment does not alter the overall conclusion of the manuscript, namely that the inhibition of the Orai channels through the compound 5D ameliorates the development of murine lung inflammation. We ourselves were surprised when we analyzed the RNA sequencing between cells with and without compound 5D to find that metabolic and mitochondrial pathways were the most dramatically affected. Further literary research demonstrated that this phenomenon had been identified to a degree during Orai inhibition in Th17 cells (PMID: 30773462). We decided to follow the data in an effort to further identify mechanistic means for ILC2 downregulation. We believe these mitochondrial mechanisms have the exciting potential to open an avenue for previously underrecognized therapeutic targets in allergic asthma.

2. For the in vivo studies, why was the *Alternaria* treatment only done for 3 days? Why did the authors choose to start the 5D treatment at the same time as the *Alternaria* treatment?

We apologize that the **Figure 6** schematic numbering is confusing. As indicated by the arrows, the *Alternaria alternata* treatment was performed for four consecutive days, per our published protocol. We chose to begin 5D treatment on day 0 with *A. alternata* so we could fully understand the effect of blocking Ca²⁺ influx on ILC2 effector function from the exact moment they are stimulated by epithelial alarmins following lung inflammation.

3. Why was the time schedule for the humanized mouse studies (2 days of treatment, Figure 7E) different from the *Alternaria* studies?

IL-33 treatment in the humanized mouse models was performed for 3 days, per our established protocol. While *Alternaria alternata* is the most physiologically relevant model for murine ILC2-dependent asthma, both mouse and human cells recognize and respond to *Alternaria alternata* stimulation. In contrast, murine cells will not recognize human IL-33. For this reason, we utilized human IL-33 in this model as a direct and specific way to stimulate the human ILC2s we adoptively transferred into the alymphoid mice, as opposed to the surrounding immune cells. We can be assured then that the response created by this model is due to the direct stimulation of the ILC2s we transferred into the alymphoid mice and does not implicate other inflammatory immune cells, such as macrophages or APCs.

4. How do the authors explain the lesser response in lung resistance in Figure 7E in comparison to both Figure 6B and G?

Unfortunately, the differences in murine asthma models (IL-33 vs. *A. alternata*) make it impossible to compare between the experiments. *Alternaria alternata*, because it is a natural fungus that induces airway inflammation as opposed to direct epithelial alarmin IL-33 stimulation, tends to produce an overall heightened pulmonary inflammatory response. This can also be seen in the number of eosinophils in the lung, comparing **Figure 6H** to **Figure 7G**.

Reviewer #3:

In this manuscript, the authors examined the roles of Orai channels, the pore components of CRAC channels, in effector functions of ILC2s. ILC2s play key roles in type 2 immunity. However, our knowledge is relatively limited regarding the mechanisms to control ILC2s. No or few studies have been performed previously to dissect the roles of Ca²⁺ signaling pathway in ILC2s. This manuscript addresses this major gap in our knowledge, and therefore, it is novel and potentially important. The experiments are straightforward by using an Orai inhibitor and ILC2 that are genetically deficient in Orai and in vitro culture and in vivo adoptive transfer models. The authors also analyzed metabolic and mitochondrial homeostasis of ILC2s to dissect the mechanisms.

The authors concluded that Orai channels play a crucial role in the functional mitochondria electron transport chain and subsequent pro-inflammatory effector function of ILC2s. This conclusion is largely supported by the experimental data. However, there are several questions remaining regarding the physiologic functions of Orai in ILC2s and specificity of the Orai inhibitor. In addition, in vivo experiments would be more convincing if a model relevant to asthma was used.

We thank the Reviewer for the constructive comments. We have now performed all the suggested experiments and believe it has further strengthened our manuscript. Please find our detailed response below

Specific comments:

1. To the best of this reviewer's knowledge, this manuscript is the first to report the expression and function of Orai channels in ILC2s. While their expression is clearly demonstrated in Figure 1, the data to support the function of Orai are rather weak as it depends on the use of a pharmacological agent thapsigargin (Figure 2). Several fundamental questions remain regarding Orai channels in ILC2s. Are Orai channels involved in more physiological events, such as oscillation of Ca²⁺ at a resting state or stimulus-dependent Ca²⁺ influx, such as that induced by leukotrienes or potentially cytokines?

This is a very interesting question. We do believe that Orai channels regulate calcium influx to a degree at baseline resting state, as previous reports have shown that loss of Orai1 results in a reduction of baseline Ca²⁺ levels in T cells (PMID: 24819389). Here, we and others demonstrate that blocking CRAC channels prior to stimulation affects effector function in various immune cells (PMID: 25938788, **Figure 2**). However, it is clear from our **Figure 1** that IL-33 stimulation does also induce Orai1 and Orai2 expression on ILC2 cells, demonstrating that the pathway also responds to alarmins. The exact pathway of this induction is unclear, but it has been previously shown that IL-33 induces the expression of STIM1 via p38 and AP1 signaling in epithelial cells (PMID: 25016017). It is feasible to hypothesize, therefore, that IL-33 stimulation upregulates Orai channel expression and calcium influx potentially via p38 and AP1. Whether this is also true for ILC2s has yet to be investigated and we are currently looking into this possibility. We hope to report these results in a future manuscript, and now include this as a future direction in our discussion.

2. Many of the experiments in this manuscript dependent on a pan-Orai inhibitor 5D. The scientific rigor of this manuscript would be enhanced with more characterization of the effects of 5D. For example, what is the dose and effect relationship between 5D and ILC2 effector function? How were the doses of 5D used for in vitro and in vivo experiments selected? Is ILC2 apoptosis or viability affected when ILC2s are cultured with 5D for 72 hours as was done in

Figure 2B through 2E? These specificity questions would be particularly important as Orai channel inhibition may alter mitochondrial health (Reference 18 of the manuscript).

This is an excellent point also raised by Reviewer 1. As mentioned above, we have now performed a new set of experiments pertaining to titration and viability now presented in **Figure S2A**. Additionally, we performed further experiments demonstrating the dose effect of compound 5D on ILC2 cytokine secretion, now found in **Figure S2C**. This experiment was designed following the schematic outlined in **Figure 2B**. Briefly, pulmonary ILC2s were FACS-sorted and cultured for 24 hours with varying concentrations of compound 5D or the vehicle. IL-33 or PBS was added to the culture for an additional 48 hours. After this time, the ILC2s were collected, and intracellular staining of IL-5 and IL-13 was assessed. Cytokine secretion is not statistically affected until the cells are cultured with 5D at a concentration of 5 μ M. At this concentration, IL-5 and IL-13 production are statistically decreased. The results of **Figure S2A** helped us determine that 5 μ M is the most effective dose in regard to decreasing cytokine production, while having the least effect on viability on the ILC2s.

3. In Figure 2F and 2G, Orai-deficient ILC2s were expanded by IL-33 and then type 2 cytokine expression was analyzed. Does Orai-deficiency affect proliferation and viability of ILC2s?

We have further performed multiple experiments addressing these questions, now found in **Figure S2G-H**. In **Figure S2G**, we demonstrate that Orai-deficiency does statistically decrease proliferation. Briefly, pulmonary ILC2s were FACS-sorted from Orai1^{-/-}, Orai1/2^{-/-} or control mice and expanded with IL-33 in vitro. Tamoxifen was then added to the cultures for 48 hours to induce deletion. The tamoxifen was washed off, and the cells were assessed for proliferation and viability. As it pertains to ILC2 viability, as seen in **Figure S2H**, we do see an expected amount of cell death after tamoxifen addition. We also see there does appear to be a slight decrease, though not statistical, in cell viability when both Orai1 and Orai2 are deleted. This is also expected to some degree, as calcium is involved in a number of pathways in the ILC2s. We however do not believe this affects other experiments in our manuscript, such as the adoptive transfer experiment, as cells were normalized after tamoxifen addition before being injected into the alymphoid mice.

4. In Figure 2G, IL-5 production was minimally affected by the deficiency in Orai1 while IL-13 production was significantly inhibited. How can the authors explain the dissociation between these type 2 cytokines?

It is important to note that IL-5 and IL-13 are regulated by different mechanisms and that there are potentially different transcription factors involved in their protein expression (PMID: 24613091, 29907525). These differences may further be explained by the fact that the effects of IL-5 and IL-13 are distinct from each other, as we and others have previously shown that ILC2 effector functions can be driven by either IL-13 or IL-5 independently (PMID: 29427641, 30755607).

5. In Figure 2, the authors concluded that “blocking Ca²⁺ entry specifically through Orai channels paly a crucial and previously underrecognized role the effector function of ILC2s”. This statement appears to be rather an over-interpretation as the authors did not study IL-33-induced Ca²⁺ entry through Orai.

We have amended our statement.

6. Similarly in Figure 4, the authors concluded that “calcium flux is indispensable for metabolism”. This statement also appears to be over-interpretation as the calcium flux was not examined. Can the calcium flux be demonstrated in the model used in Figure 4, and does 5D inhibit the flux? Does chelation of extracellular calcium or zero calcium medium (and therefore no calcium flux) have similar effects as shown in Figure 4?

We apologize, the text was supposed to read “calcium influx”, not “calcium flux”. The manuscript text has been corrected.

7. In Figure 6, The role of ILC2s was examined in vivo by acute exposure to *Alternaria*. Because the authors propose the potential therapeutic application of 5D to treat asthma, a model more relevant to asthma could be used. Can ILC2s be suppressed by 5D during chronic airway inflammation?

We have now included a chronic model of *Alternaria alternata*, as found in **Figure S4**. Briefly, mice were challenged for 3 consecutive days with 5 µg *Alternaria alternata* extracts i.n. in 50 µl. Mice were then challenged every 4 days for the following three weeks. A cohort was also given i.p. injections of 5D or vehicle. On day 26, AHR, BAL eosinophils and lung ILC2s were assessed. In confirmation of our previous findings, we also saw a statistical decrease in airway hyperreactivity, number of lung eosinophils in the BAL, as well as number of ILC2s in the lung (**S4A-D**). Our novel data therefore suggest that ILC2s can be suppressed by 5D during a chronic airway inflammation.

Minor comments:

1. In Figure 6B, the effects of 5D on *Alternaria*-induced type 2 cytokine production and airway eosinophilia were examined. Does 5D affect migration of eosinophils?

Yes, it is probable that 5D does affect eosinophil migration, though further investigation is required to determine whether 5D directly affects eosinophils or simply indirectly. We know that, indirectly, ILC2 IL-5 production is downregulated, which in turn will decrease the number of eosinophils that migrate to the lung. The direct effect of 5D on other immune requires further investigation and would be an interesting future study.

2. In Figure 6D, all the lineage-negative cells appear to be inhibited by treatment with 5D. The gating strategy for lineage-negative cells and ILC2s need to be provided.

We have now provided the gating strategies in **Figure S3**. We apologize for any confusion and have adapted the figure with more representative samples.

REVIEWERS' COMMENTS

Reviewer #1 (Remarks to the Author):

The authors have carried out several new experiments that strengthen the original manuscript considerably. All my comments have been nicely addressed. In a future study, it might be interesting to compare Synta66 with compound 5D, because Synta shows some selectivity to Orai1 over Orai2 and 3.

This is a very thorough, important and impactful study.

Reviewer #2 (Remarks to the Author):

No further questions. Thank you

Reviewer #3 (Remarks to the Author):

I have no more comments to this manuscript.

Reviewer #4 (Remarks to the Author):

This is a revised version of this manuscript that is improved. The authors have supplied crucial data on testing the pharmacological compound. I have no additional concerns.

REVIEWERS' COMMENTS

Reviewer #1 (Remarks to the Author):

The authors have carried out several new experiments that strengthen the original manuscript considerably. All my comments have been nicely addressed. In a future study, it might be interesting to compare Synta66 with compound 5D, because Synta shows some selectivity to Orai1 over Orai2 and 3.

This is a very thorough, important and impactful study.

Reviewer #2 (Remarks to the Author):

No further questions. Thank you

Reviewer #3 (Remarks to the Author):

I have no more comments to this manuscript.

Reviewer #4 (Remarks to the Author):

This is a revised version of this manuscript that is improved. The authors have supplied crucial data on testing the pharmacological compound. I have no additional concerns.

We thank all reviewers for their positive returns on our revised manuscript.